# Perception as a closed-loop convergence process

Ehud Ahissar*, Eldad Assa

Department of Neurobiology, Weizmann Institute of Science, Rehovot, Israel

**Abstract** Perception of external objects involves sensory acquisition via the relevant sensory organs. A widely-accepted assumption is that the sensory organ is the first station in a serial chain of processing circuits leading to an internal circuit in which a percept emerges. This open-loop scheme, in which the interaction between the sensory organ and the environment is not affected by its concurrent downstream neuronal processing, is strongly challenged by behavioral and anatomical data. We present here a hypothesis in which the perception of external objects is a closed-loop dynamical process encompassing loops that integrate the organism and its environment and converging towards organism-environment steady-states. We discuss the consistency of closed-loop perception (CLP) with empirical data and show that it can be synthesized in a robotic setup. Testable predictions are proposed for empirical distinction between open and closed loop schemes of perception.

## Introduction

Until the midst of the 20[th] century psychologists and psychophysicists viewed perception as a primarily active process: perception is what emerges when an organism equipped with a brain interacts with its environment (*James, 1890*; *Koffka, 1935*; *Mach, 1959*; *Merleau-Ponty, 1962*; *Uexkull, 1926*). Indeed, behavioral studies revealed that although mammals can perceive events or objects while being passive, most of the time mammalian individuals seek for objects and perceive the world via active body and sensor movements (*Ahissar and Arieli, 2001*; *Diamond et al., 2008*; *Findlay and Gilchrist, 2003*; *Halpern, 1983*; *Kleinfeld et al., 2006*; *Konig and Luksch, 1998*; *Land, 2006*; *Lederman and Klatzky, 1987*; *Najemnik and Geisler, 2005*; *Rucci et al., 2007*; *Schroeder et al., 2010*). Investigating perception at the neuronal level, however, proved to be extremely challenging and neuroscientists have adopted a series of reductionist methods in which various components of the process have been eliminated. One critical such component has been sensor motion – neuroscientists have been investing enormous efforts in precluding sensor movements as these movements, naturally, interfere with systematic characterizations of neuronal responses. This passive paradigm indeed yielded invaluable descriptions of neuronal circuits and pathways that can convey sensory information and suggested how these pathways might process sensory information. Crucially, however, passive paradigms cannot reveal how sensory information is actually processed during active perception (*Ahissar and Arieli, 2001*; *Ennis et al., 2014*; *Maravall and Diamond, 2014*). For that, a unified analysis of the motor and sensory components engaging brains with their environments is required.

Experimental data are usually examined in light of, and reflect on, implicit or explicit hypotheses. One salient outcome of the passive reductionist approach has been the over emphasis of open-loop schemes of perception. The elimination of motor components from the experimental scheme yielded a parallel elimination of motor variables from the corresponding theoretical schemes, leaving models of perception as sensory-only open-loop schemes (e.g., *Connor and Johnson, 1992*; *Edelman, 1993*; *Marr, 1982*; *Poggio and Serre, 2013*). In agreement with previous suggestions

*For correspondence: ehud.ahissar@weizmann.ac.il

**Competing interests:** The authors declare that no competing interests exist.

**eLife digest** How do we perceive the world around us? Today the dominant view in brain research is that sensory information flows from the environment to our eyes, fingers and other sense organs. The input then continues on to the brain, which generates a percept. This process is referred to as "open-loop perception" because information flows through the system predominantly in one direction: from the environment, to the sense organs, to the brain.

Open-loop perception struggles to account for a number of key phenomena. The first is that sensation is an active process. Our eyes and hands constantly move as we interact with the world, and these movements are controlled by the brain. According to Ahissar and Assa, a more accurate view of perception is that the brain triggers the movement of the sense organs, and thereby alters the sensory information that these organs receive. This information is relayed to the brain, triggering further movement of the sense organs and causing the cycle to repeat. Perception is therefore a "closed loop": information flows between the environment, sense organs and brain in a continuous loop with no clear beginning or end.

Closed-loop perception appears more consistent with anatomy and with the fact that perception is typically an incremental process. Repeated encounters with an object enable a brain to refine its previous impressions of that object. This can be achieved more easily with a 'circular' closed-loop system than with a linear open-loop one. Ahissar and Assa show that closed-loop perception can explain many of the phenomena that open-loop perception struggles to account for. This is largely because closed-loop perception considers motion to be an essential part of perception, and not an artifact that must be corrected for.

The open- and closed-loop hypotheses should now be compared systematically. One approach would be to construct an artificial perceiver (or robot) based on each hypothesis and examine its behavior. Another would be to perform experiments in which the two hypotheses make opposing predictions. Paralyzing a sensory organ without affecting the flow of sensory information, for example, would impair perception according to the closed-loop hypothesis, but would have no effect according to the open-loop hypothesis.

(*Ahissar and Arieli, 2001*; *Dewey, 1896*; *Freeman, 2001*; *Kelso, 1997*; *Port and Van Gelder, 1995*), we claim that such a reductionist paradigm should ultimately fail to elucidate neural mechanisms of natural perception. This is not to say that any reduction would fail, but to emphasize that an appropriate reductionist paradigm should leave the motor-object-sensory interactions intact. The current paper describes an attempt to bring the motor variables back to the theoretical modeling of perception, by proposing a motor-sensory closed-loop scheme for the perception of the external environment. The paper makes use of ideas previously developed in various dynamic theories (*Ahissar and Vaadia, 1990*; *Ashby, 1952*; *Kelso, 1997*; *O'Regan and Noe, 2001*; *Port and Van Gelder, 1995*; *Powers, 1973*; *Wiener, 1949*) and, in general, refers to the perception of external objects as a process of acquiring information about presently-existing external objects, whether consciously or not. The paper addresses perceptual acquisition - mechanisms of perceptual reports and their interactions with perceptual acquisition are not addressed here. It is noted, however, that a comprehensive understanding of perception depends on the understanding of report mechanisms as well. For simplicity, the term "brain" is often used in this article in an extended form that includes the sensory organs and their affiliated nerves and muscles.

## The open loop perception (OLP) doctrine

Closed loops are systems in which every signal eventually affects its source; open loops are systems in which signals cannot affect their sources. Clearly, brains contain closed-loops at all levels, some of which have been implicated in relation to perceptual processing (*Ahissar and Kleinfeld, 2003*; *Edelman, 1993*; *Martin, 2002*; *Pollen, 1999*). Yet, whether perceptual acquisition is considered an open-loop or closed-loop process does not depend on the existence of closed loops within the chain of processing, but on whether the entire chain of processing is closed (as a loop) or open. Thus, a perceptual process that starts at the sensory organ and ends somewhere in the brain, whether

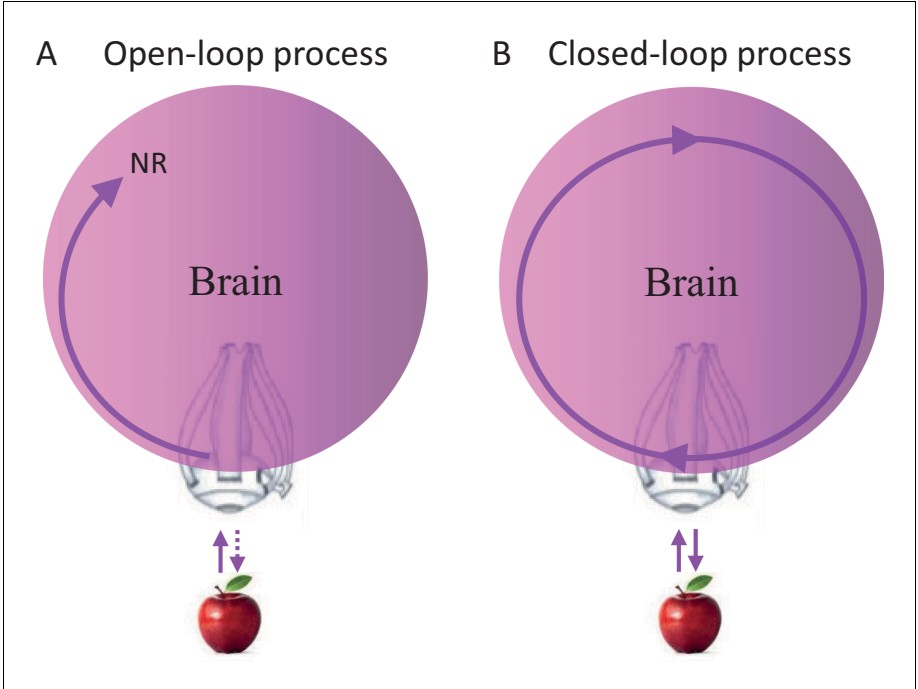

**Figure 1.** Possible perceptual schemes. (**A**) An open-loop scheme (in the motor-sensory sense) – perception begins with an interaction (uni- or bi-directional) between the object and the sensory organ (an eye in this illustration) and ends somewhere in the brain where a relevant neuronal representation (NR) is formed. (**B**) A closed-loop scheme (in the motor-sensory sense) – perception is a circular process, with no starting or ending points, which contains the sensory organ.

containing local loops or not, is termed here an open-loop perceptual (OLP) process (*Figure 1A*), whereas a perceptual process that includes the sensory organ but has no starting nor ending point, is termed a closed-loop perceptual (CLP) process (*Figure 1B*).

The OLP doctrine holds that external objects and features are perceived in an open-loop manner, in the motor-sensory sense (*Baars, 2002*; *Dehaene et al., 1998*; *Hochstein and Ahissar, 2002*; *Riesenhuber and Poggio, 2000*; *Tononi and Koch, 2008*; *Ullman, 2007*). Thus, for example, an apple activates retinal receptors, which in turn initiate a stream of activations in the brain, some of which may depend on internal loops, i.e., loops that do not include the sensory organ. An activity pattern that is repeatedly evoked in a given neuronal network in response to a presentation of the apple, and/or when such an apple is perceived, is often termed a neuronal correlate or neuronal representation (NR) of that apple. NRs are representations that are not necessarily consistent or unique, i.e., they may appear in only some of the cases in which the apple is presented or perceived, and may appear also when other objects are presented or perceived. If a specific NR is evoked in a given brain for each and every perceived appearance of the apple, is invariant to changes in internal and environmental conditions, and is unique to the apple, it can be termed "the" invariant representation (IvR) of the apple in that specific brain. Assuming OLP, IvRs should be invariant to the acquisition mode. Visual IvRs of the apple, for example, should be the same in passive and active acquisition modes, i.e., when the eye is stationary and the object moves or flashes (passive mode) and when the object is stationary and the eye moves (active mode).

The search of NRs that are also IvRs, during the last 6–7 decades, yielded several key findings. Among those is the characterization of NRs of various external features along the relevant sensory streams. For example, NRs of brief presentations of visual elements, such as dots and bars, were characterized among retinal, thalamic and cortical neurons (*Hartline, 1938*; *Hubel and Wiesel, 1962*). NRs of more complex visual patterns were characterized in various cortical areas (*Creutzfeldt and Nothdurft, 1978*; *Fujita et al., 1992*; *McMahon et al., 2014*). Crucially, however, although partial invariance had been demonstrated for portions of the proposed NRs in some of the

cases, none of these NRs was shown so far to be "the" IvR of a specific external object or feature, namely an NR that is (at least substantially) invariant to changes in the most relevant conditions of perception. Moreover, none of these studies provides information that can discriminate between OLP and alternative hypotheses. Consider, for example, studies exhibiting single neurons that increase their firing rate significantly and selectively for a given object (e.g., a face) out of several presented objects, and for several variations of that object (*McMahon et al., 2014*; *Quiroga et al., 2005*; *Viskontas et al., 2009*). The critical factor here is that such a neuron cannot be considered as describing the IvR of that object, neither as describing a reliable projection of the IvR. Based on combinatorial considerations and response variations the assumption in such cases is that the elevated firing rate of such a neuron is a (tiny) component of the relevant NR, and not the NR itself. The question is, then, would the assumed NR be invariant to a sufficiently large portion of all relevant variations of object presentation and context. Given that these neurons are not completely invariant even to the limited sample of variations presented to them (as is evident from the substantial trial-by-trial variability of their responses) and their tiny contribution to the actual NR, it is impossible to infer the level of invariance of the actual NR out of the firing patterns measured from these neurons.

Studying the passive mode of sensation also revealed various forms of internal transformations between NRs, such as, for example, transformation from NRs of static dots to NRs of static bars (*Hubel, 1996*; *Reid, 2001*), from temporal-code based NRs to rate-code based NRs (*Ahissar et al., 2000*) or from rate-code based NRs to temporal-code based NRs (*Cleland, 2010*). Clearly, these mechanisms can function within both OLP and CLP schemes of perception. Passive-mode experiments were also instrumental in describing the minimal exposure times required for generating meaningful perceptual reports. Across a large set of stimuli it was found that, depending on practice, exposure times as short as a few tens of milliseconds already allow a categorization of the presented stimulus, at least in a binary manner. As will be shown below, these findings are consistent with both OLP and CLP schemes.

## Challenges to the OLP doctrine

As described above, the OLP doctrine allowed an invaluable characterization of various components of the perceptual systems of mammals, using a set of reductionist steps. In order to verify that these specific reductions of the perceptual process are scientifically valid, one has to reconstruct perception by combining back the individual identified components. Succeeding in doing so will not only validate the specific reductionist approaches used, but, more importantly, show that OLP can be considered as a valid (i.e., self-consistent) theory of perception. At this stage we can ask whether OLP is consistent with the data collected so far. We describe here several major findings that appear to be inconsistent with OLP and thus significantly challenge the validity of OLP as a mechanism for natural perception in mammals.

### Sensation is normally active

Mammalian sensory organs usually acquire information via movements (*Ahissar and Arieli, 2001*; *Chapin and Woodward, 1982*; *Diamond et al., 2008*; *Kleinfeld et al., 2006*; *Konig and Luksch, 1998*; *Land, 2006*; *Lederman and Klatzky, 1987*; *Prescott et al., 2011*; *Rucci et al., 2007*; *Schroeder et al., 2010*). The strategies employed by sensory systems are often similar. Visual and tactile systems, for example, employ movements of sensory organs that contain two-dimensional arrays of receptors. The movements serve several functions. Larger movements (e.g., ocular saccades and head or arm movements) quickly move the array of receptors from one region of interest to another. Smaller (and slower) movements (e.g., fixational drifts and finger or vibrissal scanning) scan the region of interest at fine resolution (*Ahissar and Arieli, 2001*). This move-dwell-move pattern is typical for perceptual exploration across a large range of temporal scales, from minutes to less than a second (*Figure 2*). Olfaction and taste are probably as active as touch and vision (*Halpern, 1983*; *Kepecs et al., 2006*; *Mainland and Sobel, 2006*; *Welker, 1964*). The extent of action in hearing is less clear - while cochlear amplification is considered active (*Dallas, 1992*; *Nin et al., 2012*), whether auditory sensation is typically obtained via sensor activation is still not known (see *Perceptual systems are organized as motor-sensory-motor (MSM) loops* and *Contrasting OLP and CLP – discriminatory testable predictions* below). Cross-modal effects between body and sensor movements, which are not discussed in this paper, are likely to play a significant role in perception

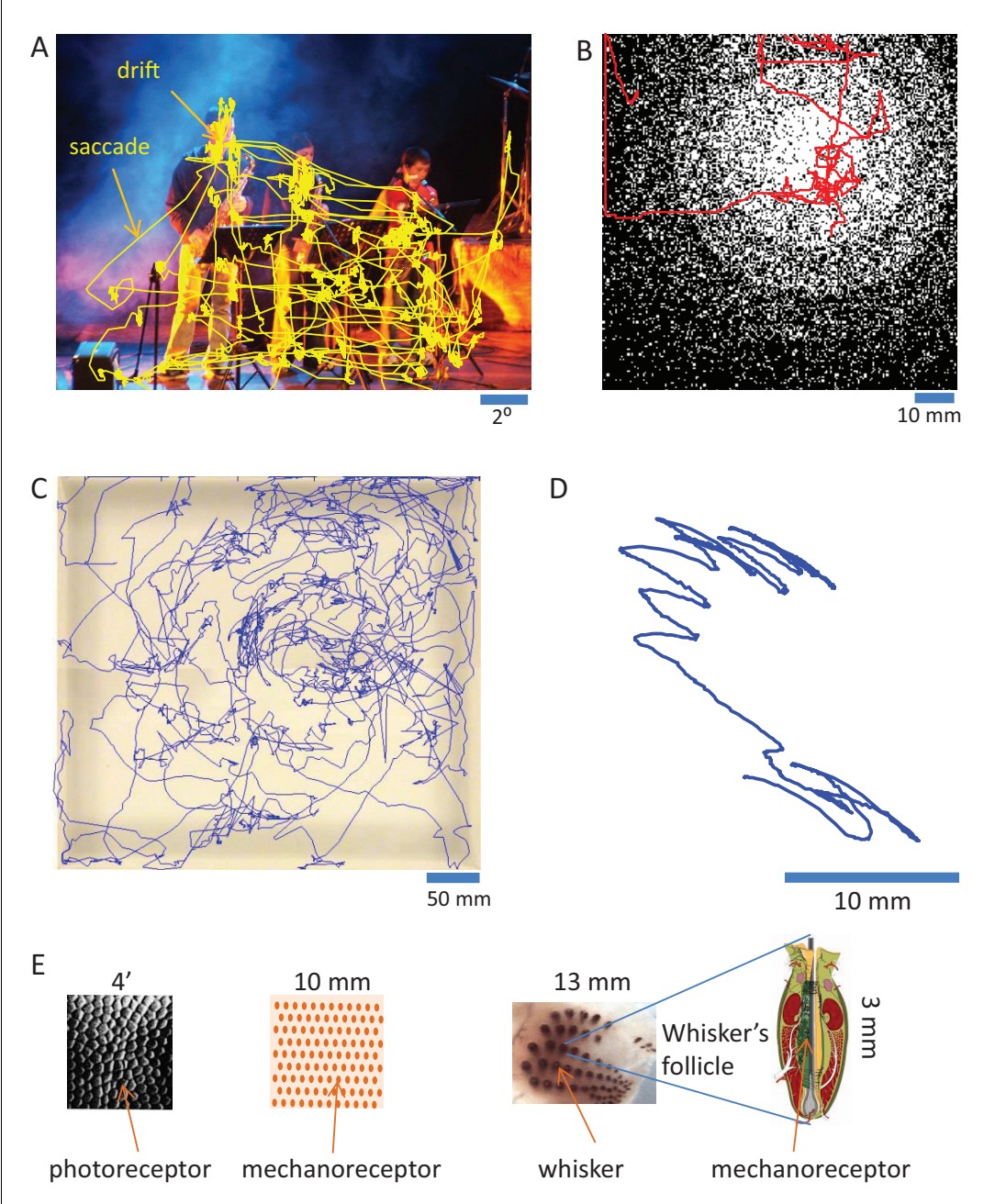

**Figure 2.** Active sensing. (A) Ocular scanning of a scene.The trajectory of a human subject's gaze (of one eye) during free viewing of an image presented on a computer screen is depicted. "Drift" points to the slow eye movements scanning a region of interest during a fixational pause. "Saccade" points to a rapid saccadic eye movement moving the gaze from one fixational pause to another. Section duration: 60 s; sampling: 240 Hz. Courtesy of Moshe Fried and Amos Arieli. (B) Manual scanning of a surface. The trajectory of a human subject's hand, while palpating a virtual surface with a varying density of elevated dots (black), is depicted. The surface was mimicked via a tactile computer mouse system (VTPlayer; VirTouch, Jerusalem) whose two 4x4 pin arrays, which were touched constantly with the index and middle fingers of the right hand, reflected the spatial details of the virtual surface according to mouse location. Section duration: 152 s; sampling: 125 Hz. Courtesy of Avraham Saig and Amos Arieli. (C) Facial scanning of an arena. The trajectory of the snout of a rat, exploring an arena using sniffing and touch, is depicted. Section duration: 828 s; sampling: 25 Hz. Courtesy of Ben Mitchinson, Chris J. Martin, Robyn A. Grant and Tony. J. Prescott; see (**Mitchinson et al., 2007**). (D) Local vibrissal scanning. The trajectory of a point near the middle of whisker C1 of a rat, exploring a region of an arena, is depicted. All whiskers except row C were trimmed on both sides of the snout. Section duration: 1.5 s; sampling: 500 Hz. Courtesy of Tess Oram, Noy Barak and Dudi Deutsch. (E) Sensory granularity. Left, a sample of retinal photoreceptors array of the human foveal area (from **Curcio et al., 1987**). Middle, a schematic illustration of the organization of one type of mechanoreceptor (rapidly adapting) under the skin of the human fingertip. Right, whiskers array: left, the array of whiskers

*Figure 2 continued on next page*

*Figure 2 continued*

across the right snout of a rat, courtesy of Sebastian Haidarliu; right, a schematic illustration of a whisker's follicle containing hundreds of mechanoreceptors, courtesy of Satomi Ebara.

as well (*Ayaz et al., 2013*; *Fonio et al., 2016*; *Grion et al., 2016*; *Keller et al., 2012*; *Moore et al., 2013*; *Niell and Stryker, 2010*). During sensor scanning, activations of individual (e.g., photo- or mechano-) receptors are functions of the interactions between the moving sensor and the physical features of external objects (*Ahissar and Arieli, 2012*; *Ahissar and Vaadia, 1990*; *Bagdasarian et al., 2013*; *Boubenec et al., 2012*; *Friston, 2010*; *Gamzu and Ahissar, 2001*; *Gibson, 1962*; *Hires et al., 2013*; *Horev et al., 2011*; *Jarvilehto, 1999*; *Kuang et al., 2012*; *Mainland and Sobel, 2006*; *O'Regan and Noe, 2001*; *Pammer et al., 2013*; *Quist and Hartmann, 2012*; *Quist et al., 2014*; *Rucci and Victor, 2015*; *Saig et al., 2012*; *Saraf-Sinik et al., 2015*; *Smear et al., 2011*). These dependencies are termed here in general motor-sensory contingencies (MS-contingencies); they form one class of the sensorimotor contingencies described by O'Regan and Noe (*O'Regan and Noe, 2001*).

The fact that mammalian sensation is active significantly challenges the OLP doctrine. First, it turns out that the common reductionist approach in which stimuli are flashed on passive sensory organs cannot be extended back to natural conditions. This is because in such experiments no information is obtained about the dependency of sensory signals on natural active interactions with the object, interactions that cannot be mimicked with passive sensors. In vibrissal touch, for example, a crucial sensory variable is the whisker curvature (*Bagdasarian et al., 2013*; *Boubenec et al., 2012*; *Quist and Hartmann, 2012*), which cannot be physically mimicked with only external forces (*Bagdasarian et al., 2013*). In vision, while the conditions accompanying an ocular drift can be mimicked, in principle, by drifting the entire visual field, the conditions accompanying ocular saccades cannot be mimicked with passive eyes. Ocular saccades are accompanied by peri-saccadic suppression during which, unlike with flashed stimuli, activity along the visual sensory pathway is significantly suppressed (*Hamker et al., 2011*). Also, saccades are always ending with additional eye movements, such as overshoots, corrections and drifts, which are lacking in passive-eye experiments. In general, it seems that the conditions introduced when stimuli are flashed on passive sensors mimic a small set of naturally-occurring states such as lightning at night or a sudden wind blowing over the rat's whiskers. It is thus not surprising that, when compared, the characteristics of NRs revealed with passive sensors are substantially different from those revealed with active sensors (e.g., *Kagan et al., 2002*).

OLP assumes that the presentation of an object retrieves the NR that represents it, i.e., its neuronal IvR. When this assumption was tested computationally at the presence of simulated eye movements it was found that such a retrieval is possible with a very simple environment (one stimulus) and a limited number of possible NRs (two), in which case the knowledge of the statistics of sensor motion (e.g., eye movements) can provide unique, unambiguous solutions (*Pitkow et al., 2007*). However, it is not clear if a similar mechanism can work with more crowded environments, even when the movement trajectory of the eye is tracked by the perceiver (*Burak et al., 2010*). The major challenge with IvR retrieval in OLP, even when the sensor trajectory is known (e.g., *Ahissar et al., 2015a*), is the instability of the sensory input. With spike-based representations and finite firing rates this instability is devastating – by the time required to construct a reliable representation the sensor may have already moved away and provide new inputs. With representations of fine visual details it had been shown that this is indeed the case (*Ahissar and Arieli, 2012*).

The realization that the visual system codes external objects differently in passive and active modes sets another major challenge to OLP. This difference can be attributed to the fact that while a passive eye that is stimulated by a flashed image can only use spatial coding to represent the image, a moving eye can use both spatial and temporal coding schemes. In fact, the temporal code appears to be much more accurate, and of higher spatial resolution, than the spatial code (*Ahissar and Arieli, 2012*; *Berry et al., 1997*; *Reich et al., 1997*). Thus an OLP theory assuming that the same IvR is retrieved with or without sensor motion must also assume that perception is based on the less accurate spatially-coded information and ignores (or corrects for) the more accurate temporally-coded information - clearly an inefficient strategy. As we will see below (in *Contrasting OLP*

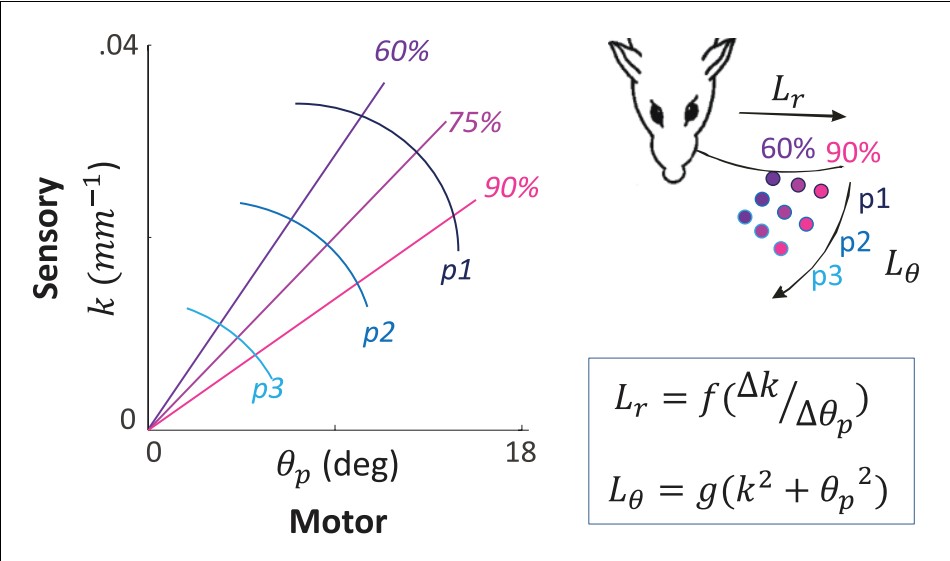

**Figure 3.** An example of MS-Contingency in vibrissal touch. A schematic illustration of morphological coding of object location (**Bagdasarian et al., 2013**) is depicted. The motor-sensory phase plane describes the combinations of values of a motor ($\theta_p$: push angle, maximal change in whisker angle from contact onset) and sensory (k, whisker base curvature) variables when a whisker actively contacts an object at various locations. The locations are defined by their coordinates in the horizontal plane (inset): three azimuth coordinates ($L_\theta$ = [p1, p2, p3]) and three radial coordinates ($L_r$ = [60%, 75%, 90%] of whisker length) are depicted and coded by colors. Note that neither of the two variables provide unambiguous coding of object location by itself; for example, k around .02 mm$^{-1}$ codes for both ~[p2, 60%] and ~[p1, 90%]. In contrast, the contingency between the motor and sensory variables provides unique coding of both $L_r$ and $L_\theta$ (see equations).

and CLP – discriminatory testable predictions), a more efficient OLP scheme, which is based on active sensing and can exploit its advantages, is also possible.

## Sensory signals convey ambiguous information

Sensory signals may often be ambiguous if processed without the motor signals that yielded them. One example is the curvature signal generated at the base of a whisker upon its contact with an object. The same curvature can be generated when contacting objects at different locations, an ambiguity that is resolved if the angle by which the whisker is rotated is taken into account (**Bagdasarian et al., 2013**) (**Figure 3**). Similarly, temporal delays between two whiskers or two photoreceptors code spatial offsets ambiguously if sensor velocity is not considered (**Ahissar and Arieli, 2001, 2012**; **Knutsen and Ahissar, 2009**). Consistently, in vivo recordings from the primate retina ruled out pure sensory processing, such as lateral inhibition, as a basis for edge detection while supporting motor-sensory processes involving eye movements (**Ennis et al., 2014**). These pieces of evidence join a substantial list of evidence for the ambiguity of sensory signals and the unambiguity of MS-contingencies (**Bompas and O'Regan, 2006**; **O'Regan and Noe, 2001**). For vision this is further supported by a series of experiments and analyses indicating that retinal information depends on the nature and trajectory of miniature eye movements (**Ahissar et al., 2014**; **Ko et al., 2010**; **Kuang et al., 2012**; **Olveczky et al., 2003**; **Rucci et al., 2007**; **Snodderly et al., 2001**).

Note that this challenge cannot be alleviated by adding efference copy information to open-loop perceptual processing – efference copies are not accurate enough to account for perceptual accuracy (**Ahissar et al., 2015a**; **Pitkow et al., 2007**; **Simony et al., 2008**). For example, perception of object location in rats (**Knutsen and Ahissar, 2009**) depends on the details of the motor trajectory at a resolution corresponding to movements induced by individual motor spikes (**Herfst and Brecht, 2008**; **Simony et al., 2008**), a resolution that is likely not available in internal efference copies (**Fee et al., 1997**; **Hill et al., 2011**). Similarly, the accuracy of visual efference copies is two orders of magnitude lower than the size of fine eye movements (**Pitkow et al., 2007**).

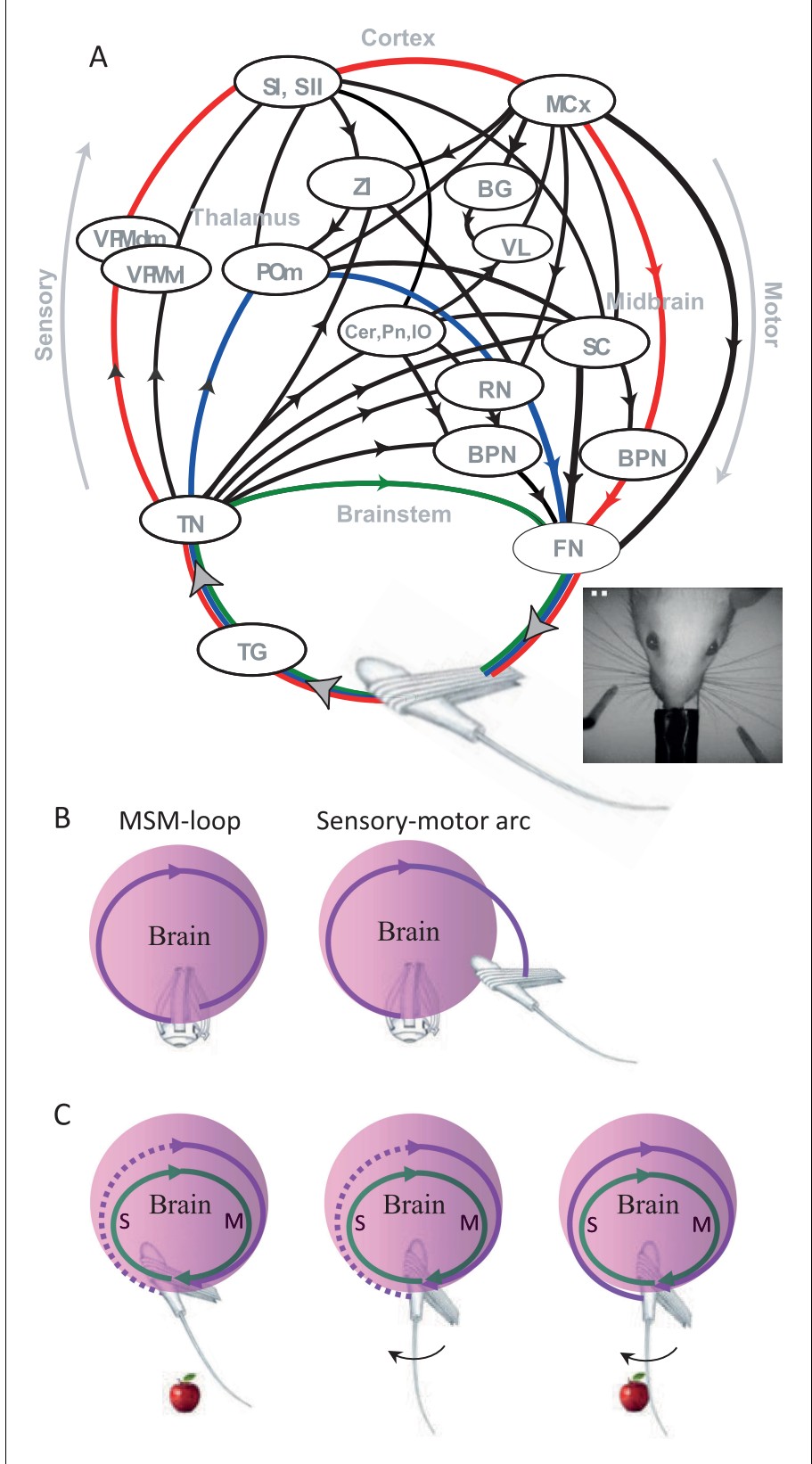

**Figure 4.** Anatomy and perceptual schemes of a sensory modality. (**A**) Closed-loop motor-sensory-motor (MSM) connections of the vibrissal system.A schematic diagram of the most relevant connections, through which sensory activities feed motor circuits at various levels, is depicted; efference copies are not explicitly depicted. Oval circles

*Figure 4 continued on next page*

*Figure 4 continued*

indicate brain regions [BPN, brainstem premotor nuclei (arbitrarily divided into two oval circles); BG, basal ganglia; Cer, cerebellum; FN, facial nucleus; MCx, motor cortex; POm, posteromedial thalamic nucleus; RN, red nucleus; SC, superior colliculus; SI, primary somatosensory cortex; SII, secondary somatosensory cortex; TG, trigeminal ganglion; TN, trigeminal brainstem nuclei; VL, ventrolateral thalamic nucleus; VPM, ventroposteromedial thalamic nucleus; ZI, zona incerta]. Black curves connecting brain regions indicate anatomical connections. Arrows indicate the direction of information flow between brain regions. Connections not labeled with arrows are reciprocal (for more details see *Bosman et al., 2011*; *Diamond et al., 2008*; *Kleinfeld et al., 2006*). Three examples of individual MSM-loops are illustrated by green (a brainstem loop), blue (a thalamic loop) and red (a cortical loop); the primary efferents (FN to muscles) and afferents (follicle to TN) may or may not be common to different pathways. Modified from (*Ahissar and Knutsen, 2008*; *Ahissar et al., 2015b*). Inset, top view of the head and whiskers of a rat performing a bilateral localization task. (B) An MSM-loop (left) activates and senses the same organ. Sensory-motor arcs (right), which sense one organ and activate another, are not discussed in this paper. (C) Inclusion in an MSM-loop. Re-afferent loops (green) are always closed and thus can be considered as constantly 'perceiving' their organs. Ex-afferent loops (magenta) are normally open (dotted). An ex-afferent loop is closed (solid) only when the sensory organ interacts with the object (right); neither object presence alone (left) nor sensor movement alone (middle) close the loop.

## Perceptual systems are organized as motor-sensory-motor (MSM) loops

Sensory organs (eyes, hands, whiskers) are associated with muscles whose activations move the sensory organ and induce sensory signals (*Simony et al., 2008*). The neuronal motor and sensory systems that are associated with a given sensory organ are connected via an intricate system of loops that does not allow an isolated operation of either (see illustration of the vibrissal system in *Figure 4A*). When motor efferents of a specific sensory organ are activated, sensory signals are inevitably generated (*Hentschke et al., 2006*; *Jarvilehto, 1999*; *Johansson and Flanagan, 2009*; *Keller et al., 2012*; *Poulet and Petersen, 2008*) and when sensory signals are generated, motor efferents to the same sensory organ are naturally affected (*Bonneh et al., 2013*; *Gilad et al., 2014*; *Ko et al., 2010*; *Mainland and Sobel, 2006*; *Nguyen and Kleinfeld, 2005*). One needs to anesthetize the brain, eliminate specific pathways, or prevent the movements of the relevant sensory organs in order to 'open' this motor-to-sensory-to-motor loop.

Brain loops that include the relevant sensory organ for a given perception (*Ahissar and Arieli, 2012*; *Ahissar and Kleinfeld, 2003*; *Ahissar and Vaadia, 1990*; *Diamond et al., 2008*; *Kleinfeld et al., 2006*; *Saig et al., 2012*) are termed here motor-sensory-motor loops, or briefly *MSM-loops*. For example, vibrissal MSM-loops include loops running via brainstem stations, thalamic stations and cortical stations, all sharing the same sensory organ (*Figure 4A*; colored arcs). Finger-touch MSM-loops include loops that are similar to those of the vibrissal system, running through homologous stations (*Ahissar et al., 2015b*). Existing anatomical descriptions of Visual MSM-loops are less detailed, although it is known that they also follow a multi-pathway architecture (*Bishop, 1959*; *Casagrande, 1994*; *Diamond, 1983*; *Lappe et al., 1999*; *Nassi and Callaway, 2009*; *Wang et al., 2007*), with sensory information feeding back onto oculomotor pathways at virtually all brain levels (*Dhande and Huberman, 2014*; *Fries et al., 1985*; *Guillery, 2005*; *Guillery and Sherman, 2002*; *Krauzlis and Lisberger, 1991*; *Malik et al., 2015*). Likewise, sniffing and tasting are likely to be controlled via modality specific MSM-loops as well (*Kareken et al., 2004*; *Kepecs et al., 2007*; *Moore et al., 2013*). As for the auditory system, relevant MSM-loops are likely those whose motor efferents activate the outer hair cells in the cochlea, which in turn change the tuning of the basilar membrane (*Guinan, 1996*; *Jennings and Strickland, 2012*), those which activate the muscles of the middle ear (*Kobler et al., 1992*) and those which control the direction of the pinnae. MSM-loops that control head movements can be shared by all cranial senses.

Throughout this paper, when we refer to MSM-loops we refer both to their anatomy and function. We use the term "motor-sensory-motor" instead of the common term "sensory-motor" in order to emphasize the fact that the loops that we refer to are those controlling a single sensory organ, and in which the flow of information is from the sensory organ to itself, via the brain. These loops should be distinguished from multi-modal sensory-motor loops, which include sensory-motor arcs that link different modalities (e.g., eye – hand or eye – whisker; *Figure 4B*) – these inter-modal loops and arcs are not addressed here.

The closed-loop architecture of the perceptual systems challenges the OLP doctrine. How would an open-loop mechanism emerge, and how would it function, in such a closed-loop system? In natural conditions every sensory activity will affect the movement of the sensory organ and evoke new sensory activations, assuming that the external object does not disappear after its first interaction with the brain. As loop cycle times are typically shorter than the typical perceptual epoch (e.g., *Deutsch et al., 2012*), a sequence of such sensory activations is typically expected within each perceptual epoch (i.e., a period of continuous engagement with the object). How would this sequence of activations be ignored? And, more importantly perhaps, why would it be ignored? Moreover, it is known that increased stimulus exposure durations increase perceptual accuracy and confidence (*Packer and Williams, 1992*; *Saig et al., 2012*); if this is achieved in an open-loop manner, then it would mean that the brain does use those additional sensory signals, and "corrects for" the motion that evoked them using efference copy signals. Unfortunately, as mentioned above (in *Sensory signals convey ambiguous information*), efference copy signals are not accurate enough to account for fine perception.

## Perception can be masked "backwardly"

Although the loops are anatomically closed, they can be opened functionally. For example, projecting a flash of an image on the retina or skin, for a duration that is shorter than the duration of the minimal MSM-loop cycle, does not allow closure of the loop. When such a 'virtual knife' is used, the system is forced to function in an open-loop mode, regardless of its architecture. According to the OLP doctrine, this reductionist step does not interfere with the fundamental process underlying perception and thus the natural perceptual process can be reconstructed from such individual open-loop processes. However, backward masking, a robust perceptual phenomenon, challenges this assumption. The presentation of a second object within tens of milliseconds after the presentation of a target object prevents or impairs the perception of this target object (*Enns and Di Lollo, 2000*). Such "backward in time" effect can occur in some open loop scenarios, for example if perception would depend on the integration of two processes, one fast and one slow, such that the fast process activated by the mask would interfere with the slow process activated by the target (*Breitmeyer and Ogmen, 2000*). Experimental data, however, were found to be inconsistent with such open loop schemes, while supporting a dependency of perception on closed loop ("re-entrant") mechanisms, in which the stimulus is repetitively sampled (*Enns and Di Lollo, 2000*). The dependency on repetitive sampling strongly challenges the assumption that the 'virtual knife' does not interfere with the natural process of perception. Backward masking indicates that flashed stimuli allow, at best, an examination of the first step of a perceptual process, as explained below (see *CLP propositions*).

## Perception involves motor-sensory convergence

Perception takes time – typical perceptual epochs last hundreds of milliseconds. The first wave of sensory-driven neuronal activity typically reaches most of the relevant cortical areas within ~100 milliseconds, and quick saccadic reports on the crude category of the perceived item can be generated as fast as 150 milliseconds after stimulus onset (*Wu et al., 2014*). Yet, the identification of more delicate categories and the perception of item details take typically hundreds of milliseconds from first sensor-object encounter, a period during which perceptual acuity continuously improves (*Micheyl et al., 2012*; *Packer and Williams, 1992*; *Saig et al., 2012*). Consistently, scalp EEG recordings reveal that perceptual thresholds are correlated with neuronal activities that are recorded after the first transient neuronal response (*Censor et al., 2009*).

Careful analyses of rodent and human behavior during tactile perception reveal signatures of a converging process. Object features, such as location and texture, are perceived via a sequence of sensor-object interactions whose motor and sensory variables show a pattern of convergence towards asymptotic values (*Chen et al., 2015*; *Horev et al., 2011*; *Knutsen et al., 2006*; *McDermott et al., 2013*; *Saig et al., 2012*; *Saraf-Sinik et al., 2015*; *Voigts et al., 2015*). This behavior is consistent with previous descriptions of perception as a dynamic process (*Ahissar and Vaadia, 1990*; *Ashby, 1952*; *Kelso, 1997*; *O'Regan and Noe, 2001*; *Port and Van Gelder, 1995*; *Powers, 1973*; *Wiener, 1949*), but not with an open-loop one. Converging dynamics, i.e., dynamics during which the state of the entire system gradually approaches a steady state, are hallmarks of

closed-loops – an open-loop system does not converge as a whole. Thus, while the OLP doctrine could accept neuronal convergence in local circuits, it cannot account for perceptually-relevant MSM converging dynamics.

## Hypothesis and Results

### The closed-loop perception (CLP) hypothesis

Here we propose a closed-loop scheme of perceptual acquisition, and suggest to refer to it as a possible alternative to the OLP doctrine. Within the scope of this paper we describe the acquisition of information about the organism's immediate environment and do not address the interactions between perceptual acquisition and perceptual report. The CLP scheme is consistent with the same data challenging OLP, primarily because it considers sensor motion as an integral part of perception rather than as a factor that needs to be corrected for. We propose to continue comparing the two alternative schemes on equal grounds against accumulating data, and for aiding such a comparison we list potentially discriminative experiments towards the end of this article.

#### The CLP hypothesis is based on the following assumptions

i. Sensation is normally active. Sensory organs obtain information about external objects via active interactions with the physical attributes of the object.

ii. MSM-loops are fundamental units of mammalian perception. These loops, as every closed loop, can approach lag-less, steady states. During steady-states all changes in the loop are fully predictable and the loop functions as one unit, with no beginning or end and with no causal order; changes in one component of the loop cannot be considered as lagging or leading changes in any other component of the loop.

iii. There are two basic types of MSM-loops (*Figure 4C*, left). The first uses proprioceptive (re-afferent; *Figure 4C*, green) signals to monitor sensor state. Such loops are always closed; that is, information about the sensor state is always conveyed back to the rest of the loop. Importantly, these loops can also sense external features in a rough way (*Berryman et al., 2006*), probably via sensing significant deviations between intended and actual sensor kinematics. The other type uses sensory signals to directly monitor features of external objects (ex-afferent; *Figure 4C*, magenta). The receptors of "ex-afferent loops", i.e., loops that contain ex-afferents, do not respond to sensor movement per-se, but to sensor interactions with external objects. These loops remain open if no object exists in the external field scanned by the sensor. They will be closed (i.e., meaningful neuronal activity will flow along the loop) only through interactions of the sensory organ with specific external features to which their receptors are responsive. For example, whisker contacts with external objects activate a family of vibrissal mechanoreceptors that otherwise would remain silent ("Touch cells") (*Knutsen and Ahissar, 2009*; *Szwed et al., 2003*) - the loops containing these neurons will be closed only through the interaction of whiskers with an external object present in the field of whisking (*Figure 4C*). Similarly, most photoreceptors are activated by luminance changes and thus would remain silent when the eye rotates against a uniform background. Visual ex-afferent MSM-loops are thus likely to be closed only via the existence of specific optical features in the visual field.

#### CLP propositions

The following set of propositions is consistent with our assumptions and defines a hypothesis for perception.

I. *Perception (of external feature(s)) ≡ a process of inclusion in MSM-loop(s)*. During this process the entire MSM-loop, including its muscles, receptors and neurons, and with the external feature being included, converges towards a steady-state. Had the loop, with the external feature included, reached steady-state, that feature could be considered as been "directly perceived" by the loop, with no mediation and no delay. However, as such a steady-state is an idealized state in which nothing new is perceived, MSM-loops never reach the absolute steady-states. Rather, they rove dynamically between being perturbed (by external or internal processes) and approaching steady-states.

Our hypothesis thus asserts that a given percept is associated with a given steady-state of the motor-sensory-neuronal variables space. This steady-state can be referred to as the IvR of the

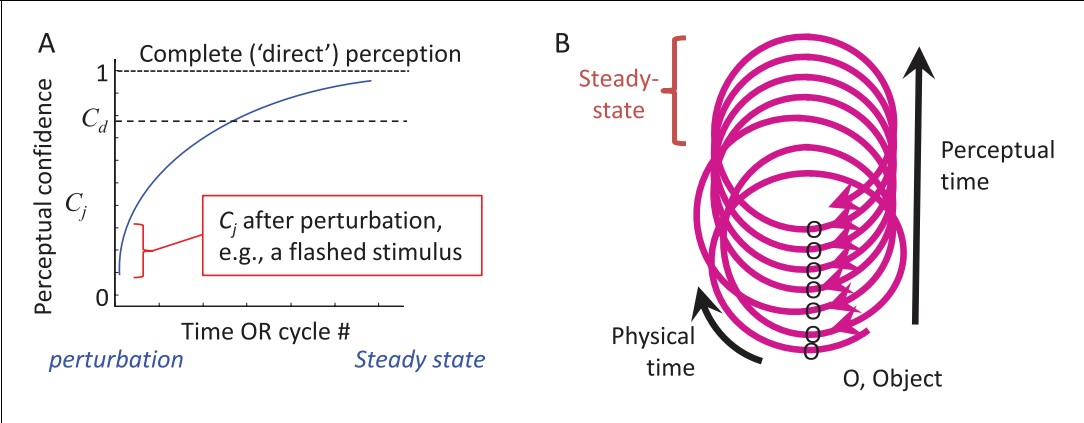

**Figure 5.** CLP dynamics. (**A**) The dynamics of perception of an individual feature by an individual MSM-loop follows a convergence pattern. The loop starts converging towards its steady-state (in which state perception is complete and "direct") upon the first interaction with the object, whether active or passive (e.g., a flashed stimulus). The confidence of perceiving feature $j$ ($C_j$) gradually increases during convergence. The loop may quit the process when $C_j$ becomes larger than a certain internal threshold ($C_d$) or upon an internal or external perturbation. (**B**) The relationships between physical and perceptual time during CLP convergence are presented via a spiral metaphor, in which the physical time can be measured along the spiral, and the perceptual time can be measured across the spiral, e.g., by counting the number of activations of a given point along the loop. A steady-state can be reached at some point along the process.

relevant feature or object. The steady-states can be of various types: a fixed point in the motor-sensory-neuronal space, a closed trajectory within this space (limit-cycle) or a chaotic attractor. We name these attractors *perceptual attractors* (***Freeman, 2001***; ***Kelso, 1997***; ***Port and Van Gelder, 1995***) since perceiving according to our hypothesis is equivalent to converging towards one such specific attractor in the relevant motor-sensory-neuronal space. A crucial aspect of such an attractor is that the dynamics leading to it encompass the entire relevant MSM-loop and thus depend on the function transferring sensor motion into receptors activation; this transfer function describes the perceived object or feature via its physical interactions with sensor motion. Thus, 'memories' stored in such perceptual attractors are stored in brain-world interactions, rather than in brain internal representations (see also ***Dreyfus, 2002***; ***Merleau-Ponty, 1962***; ***O'Regan, 1992***).

During the dynamic convergence process the state of the entire MSM-loop (with the external feature included) gradually approaches a steady-state. This can be illustrated by the dynamics of an internal variable, termed here "perceptual confidence" ($C_j$, where $j$ indicates the perceived feature), whose maximal value is obtained at steady-state (***Figure 5A***). $C_j$ starts to build up upon the first interaction with the object and gradually increases towards the steady-state asymptote as additional interactions occur (see thalamo-cortical correlates of such a process in ***Figure 6*** of ***Yu et al., 2015***). This convergence process allows for partial perception (e.g., binary classification) to occur even with very brief presentations of external stimuli (***VanRullen and Thorpe, 2001***) (***Figure 5A***, red mark).

According to CLP, thus, artificially flashed stimuli initiate a perceptual process, and provide some perceptual information, but do not allow further accumulation of perceptual information as would normally occur with natural stationary objects (***Figure 5A***). CLP thus predicts that, although the percepts evoked by flashed stimuli can be robust, they would typically include significantly less information than the information actively acquired from continuously-present objects during typical perceptual epochs. Within the CLP scheme, psychophysical data obtained with flashed stimuli are valuable for assessing the degree of convergence that can be reached upon a single interaction with the object, and its reportable resolution.

We leave the details of the generation of the confidence signal, $C_j$, outside the scope of this article. Yet, for the sake of clarity, we outline here one possible mechanism, which is based on internal models (***Anderson et al., 2012***; ***Gordon and Ahissar, 2012***; ***Kawato, 1999***; ***Lalazar and Vaadia, 2008***; ***Nijhawan and Kirschfeld, 2003***; ***Wolpert et al., 1998***). Internal models implement simulations of the interactions of the brain with the external world, simulations that are tightly coupled to the actual interactions. A continuous comparison of the predictions of internal models with the signals resulting from the actual interactions can provide a measure of the deviation of the actual

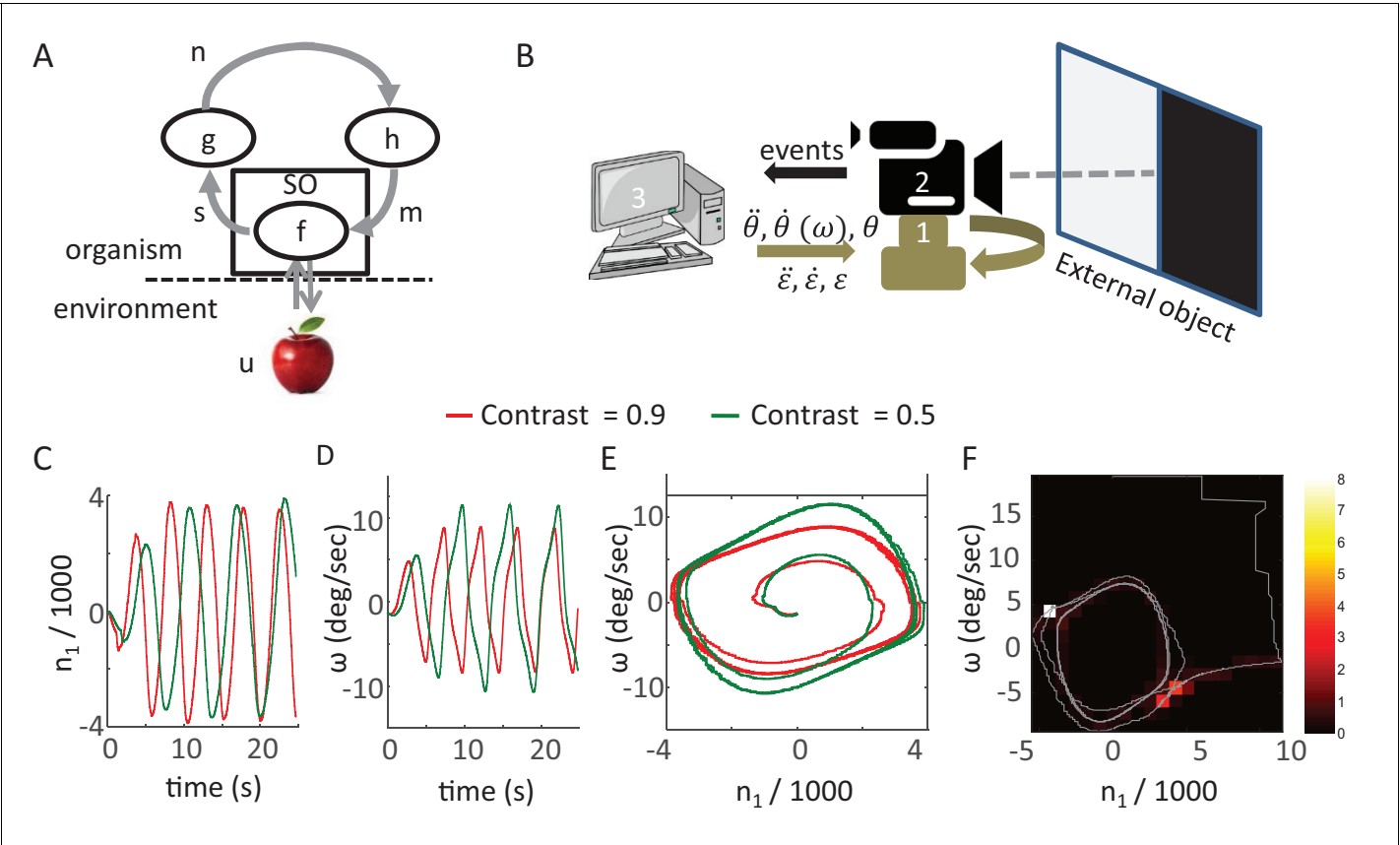

**Figure 6.** Synthesis of closed-loop perception in a robotic setup. (**A**) A sketch of the MSM-loop model template.m, motor variable; SO, sensory organ; s, sensory variable; n, neuronal variable; h, f, g, transfer functions; u the environment dynamics. The arrows depict the direction of information flow within the loop. (**B**) The SYCLOP robotic platform. A sketch of the robot with its different components: Pan-Tilt control unit (PTCU, only the pan axis was used here) (1), DVS camera (2), and desktop computer (3). The computer sends commands to the PTCU which controls the camera's rotations in the azimuth (θ) and elevation (ε) axes. The DVS camera sends visual 'on' and 'off' events to the computer. (**C-F**) Implementation of a specific contrast perceiving CLP model (see text). (**C,D**) $n_1$, the integrated difference between 'ON' and 'OFF' events, and ω, sensor angular velocity along the pan axis, (C and D, respectively) as a function of time in two different runs of the CLP algorithm, one facing a contrast of 0.9 (red) and one facing a contrast of 0.5 (green). $n_1$ is scaled in units of 1000 events. (**E**) System's trajectories in the 2D $n_1$-ω state space. Same data as in C and D. (**F**) Example of emergent smooth-pursuit like behavior when using a moving edge as a stimulus. The trajectory of the system in the $n_1$-ω plane (gray line) overlaid on a heat map where the color of each segment corresponds to the amount of time in seconds the system spent within this segment. The smooth pursuit periods are represented by the white and light red squares. While in a smooth pursuit, the camera was moving with a constant angular velocity – smoothly tracking the edge.

convergence process from an expected one - the closer the actual and simulated processes the higher the confidence. If internal models are also continuously updated along with the developing history of the organism, as usually assumed, they can provide a close estimation of $C_j$. Internal models are often hypothesized to be implemented via cerebro-cerebellar, basal-ganglia, or thalamo-cortical loops; in principle, internal models affiliated with different MSM-loops can be implemented via different brain areas or circuits. Furthermore, the internal models are likely active players in the operation of the MSM-loop and its convergence dynamics, which is consistent with reports of neuronal signals that are involved in both perceptual processing and perceptual confidence (*Fetsch et al., 2014*).

The converging process is expected to end by another external perturbation, by reaching a certain level of $C_j$ (as in bounded evidence accumulation, *Shadlen and Kiani, 2013*), by the passage of a certain time interval or by an overriding or coordinated operation of another MSM-loop. The guiding principle of brain-object disengagement, when controlled by the brain, is likely to be based on information gain – when subsequent interactions are expected to provide relatively little relevant information, the brain would typically detach from the perceived feature or object and orient its

MSM-loops towards other features or objects (*Creutzig et al., 2009*; *Horev et al., 2011*; *Little and Sommer, 2013*; *Polani, 2009*; *Saig et al., 2012*). In principle, modeling of loop disengagement can follow the modeling of decision making dynamics (*Gold and Shadlen, 2007*; *Shadlen and Kiani, 2013*) and dynamic perception (*Kelso, 1997*), targeted to entire MSM-loops rather than to local circuits and assuming active, self-induced sampling of evidence.

II. *Perception of an external object ≡ a coordinated process of inclusion in a collection of MSM-loops.* An individual MSM-loop is assumed here to typically perceive an individual feature. An 'object' is a certain set of such features, proposed here to be a set that is delineated by a coordinated convergence process. As the dynamics of such multiple-loop convergence are beyond the scope of this article, we would only mention that they should depend on two major processes. One is a binding process in which the loops share information - one candidate vehicle for inter-loop binding is a link established by fast frequency oscillations (*Fries et al., 2007*; *Tallon-Baudry and Bertrand, 1999*), as they allow several inter-loop iterations per each motor-sensory-motor iteration. The second is a selection process (*Humphries et al., 2007*; *Prescott et al., 2006*) that determines the control over the sensory organ. This selection process is not unique to 'within object' loops – it should operate constantly, as naturally more than one MSM-loop is expected to be functional at any given time. We consider two, not mutually exclusive, major schemes of control selection. In one, every loop controls a sub-set of the muscle units attached to the sensory organ (e.g., *Takatoh et al., 2013*). In the other, there is a dynamic selection of the MSM-loop(s) that control sensor-object interactions at any given moment. This process can be implemented by a variety of architectures, including subsumption-like (*Brooks, 1986*), hierarchical curiosity (*Gordon and Ahissar, 2012*; *Gordon et al., 2013*; *Gordon et al., 2014*) and others (*Arkin, 1998*). The binding between the loops is expected to break at the end of the perceptual epoch, upon the disengagement of one or more of the loops from their external features.

Hierarchical dynamics of MSM-loops can be illustrated by considering a visual scanning of an object or a scene or a tactile scanning of a surface (*Figure 2*). For example, when looking at an object or a scene the eyes saccade through a sequence of fixation areas, following a trajectory that is often termed "scanpath" (*Ko et al., 2010*; *Noton and Stark, 1971*; *Walker-Smith et al., 1977*; *Yarbus, 1967*), and drift around within each fixation area for several hundreds of milliseconds (*Ahissar et al., 2014*; *Rucci et al., 2007*; *Steinman and Levinson, 1990*). The scanpath trajectory, which moves the visual gaze from one region of interest to another, is considered in our scheme to be part of converging dynamics in one level of MSM-loops, and the local drift scanning trajectories, which acquire local visual details (*Ennis et al., 2014*), are considered to be parts of converging dynamics of MSM-loops at lower levels (*Ahissar et al., 2014*). Moving on from a given fixation area depends on the $C_j$s obtained at that area by the lower loops, on the perceptual dynamics of the scanpath loop, on variables of still higher loops depending on the context, task and brain state and on changes in the external object or scene.

III. *Perceptual time is determined by the MSM-loop's cycle time.* Physical time is unlikely to have a neuronal metric, or 'yardstick,' enabling its direct measurement. In contrast, a yardstick that is available for each MSM-loop is its own cycle time, which can be sensed by each of its components. Durations of external events can be measured by the counts of such 'ticks' (*Ahissar, 1998*). In this case, the resolution of perceptual time is the loop cycle time; events occurring within one cycle are considered simultaneous (*Poppel, 2004*). A possible relationship between physical and perceptual times can be described using a helix metaphor (*Figure 5B*). The helix should be considered flexible in its 'perceptual axis', being affected by the state of the perceiving loop. As changes in the loop's cycle time can also be sensed by neurons (*Ahissar, 1998*; *Ahissar and Vaadia, 1990*; *Buonomano and Merzenich, 1995*), online calibration between perceptual and physical time is possible to some extent. The assessment of physical time by an MSM-loop is predicted here to depend on the loop cycle time, which of course can change according to the perceptual scenario.

## Corollaries of the CLP hypothesis
Major corollaries of the CLP propositions are:

i.   An individual MSM-loop is the elemental unit of perception, namely is both necessary and sufficient for perception (of at least one external feature) to emerge in natural conditions. Thus, any reductionist study of perception must include at least one MSM-loop.
ii.  Nested MSM-loops can present different dynamics simultaneously. A higher-order loop can perceive (i.e., include) a scene at (close to) a steady-state, while lower-order loops dynamically rove along their perturbed – steady-state axis. Thus, an environment (e.g., a room) can be perceived in (close to) a "direct" manner by higher loops for the entire period in which its details are sequentially scanned by lower-order loops.
iii. Perception is a continuous dynamic and interactive process and not a momentary event (*Cleeremans and Sarrazin, 2007*; *Edelman and Tononi, 2001*); during the perceptual process, a percept gradually emerges.
iv.  Perception is associated with changes in brain dynamics rather than with the construction of invariant internal representations. Given that sensor movements are never identical, and in fact vary significantly between perceptual epochs even when objects and contexts are constant (e.g., *Knutsen et al., 2006*; *Saraf-Sinik et al., 2015*), what remain invariant are the relationships between the variables of the entire MSM-loop(s)(see also *Merleau-Ponty, 1962*; *O'Regan and Noe, 2001*). Individual neuronal variables anywhere in the brain are unlikely to remain invariant (*Rokni et al., 2007*).
v.   Perceptual time is determined by the dynamics of the relevant MSM-loops and thus depends on the perceived environment. Also, within one loop cycle period, changes in external features and internal processes occur at different physical times but at the same perceptual time.
vi.  Perception is not necessarily conscious. The brain can perceive external features by loops that are not accessible, at that moment, to conscious report. Thus, conscious perception is defined here as one category of perception.

## CLP mathematical framework and models

One natural choice of a mathematical framework for CLP is the framework of dynamical systems (*Kelso, 1997*; *Port and Van Gelder, 1995*). Within this framework each MSM-loop is modeled as a dynamical system that includes motor, sensory and neuronal variables, as well as the differential equations which describe their relations. The following is a general mathematical description of such a model (*Figure 6A*):

$$\bar{s} = f(\bar{m}, u)$$
$$\dot{\bar{n}} = g(\bar{n}, \bar{s})$$
$$\dot{\bar{m}} = h(\bar{m}, \bar{n})$$

The bars above the letters indicate that they represent a vector (of one variable or more). *g* and *h* are functions describing the intrinsic dynamics of the variables ($\bar{n}$ and $\bar{m}$ respectively) and their dependency on the variables in the preceding stations of the loop ($\bar{s}$ and $\bar{n}$ respectively). The sensory variables ($\bar{s}$) do not depend on their intrinsic dynamics in this formalization, which assumes short sensory time constants; they are determined by the motor variables ($\bar{m}$) and the state of the environment (*u*), according to the function *f*. The function *f* encapsulates the physical laws governing the sensory organ-environment interactions and the transduction of physical signals to neuronal ones.

The state of the system is defined as the vector containing all the variables ($\bar{m}, \bar{s}, \bar{n}$). Perception is achieved through the convergence of the system to a steady-state within this state-space. The information of the perceived feature is contained in the values of the dynamic variables (the system's state) at this steady-state. High-level functions such as integration of the general context or a report mechanism are not included in this model of single-feature acquisition.

## Synthesis of CLP in a robotic setup

One way to test such CLP models and demonstrate their basic behavior is to implement them using a synthetic agent. We built a simple robot for this purpose; the robot (SYCLOP: SYnthetic Closed-LOop Perceiver) includes two motors, one sensor and their bilateral connections (*Figure 6B*). This platform allows the implementation of minimal MSM-loops based models (one motor DOF and one sensor). The SYCLOP uses a biomimetic camera (DVS128, iniLabs Ltd Zurich, Switzerland, *Lichtsteiner et al., 2008*) as its sensor; this camera, like a retina, sends signals only upon luminance

intensity changes. The camera is mounted on a pan-tilt control unit (PTU-46-17, DirectedPerception, CA, USA). The motor-to-sensory connection is implemented by moving the camera along the pan-tilt axes while the sensory-to-motor connection is implemented by a computer that implements the model's equations.

The SYCLOP platform was used, for example, to implement and test the behavior of a single MSM-loop model which was designed to perceive a visual contrast. The stimulus, in this case, was presented on a computer screen: half of the screen was kept dark and on the other half a uniform grayscale surface was displayed. The grayscale values ranged from dark to white. We defined two sensory variables $r_{on}$ and $r_{off}$ - the rate of 'ON' events (single-pixel events in which the luminance intensity increased) and the rate of 'OFF' events (single-pixel events in which the luminance intensity decreased) - integrated over the entire camera's field. The characterization of the dependency of these two sensory variables on the chosen motor variable (sensor angular velocity along the pan axis, $\omega$) and the external feature (contrast, $\gamma$) resulted in the following equation:

$$r_{on} - r_{off} = C_1 \gamma \omega$$

Where $C_1$ represents a constant and noise is ignored. The MSM-loop model is completed by the addition of two transfer functions that define two differential equations, sensory-to-neuronal (g) and neuronal-to-motor (h):

$$\begin{cases} \dot{n}_1 = g(n_1, r_{on}, r_{off}) = C_2(r_{on} - r_{off}) = C_2 C_1 \gamma \omega \\ \dot{w} = h(\omega, n_1) = \frac{1}{C_3}(\mu(1 - C_4 n_1^2)C_3 \omega - n_1) \end{cases}$$

Where $n_1$ is defined as the (single) neuronal variable, which integrates the difference between $r_{on}$ and $r_{off}$ (**Demb and Singer, 2012**), $\omega$ is the (single) motor variable defined as the sensor's angular velocity and $C_2$, $C_3$ and $C_4$ represent constants. The functions g and h were chosen such that the resulting dynamical system would be equivalent (up to constants multiplications, assuming all constants and parameters are positive) to a Van der Pol oscillator (**Kanamaru, 2007**). This specific system was chosen due to its known dynamics: the system converges to a single closed trajectory within its 2D phase plane (i.e. a limit cycle) independently of the initial values of the variables. After convergence each of the dynamic variables is a periodic function of time (e.g., **Figure 6C and D**). Clearly, other dynamical systems could fit as well.

This model was implemented on the SYCLOP platform with the aid of a c program running on the computer incorporated in the platform (**Figure 6B**, item 3). The program received the ON and OFF events from the DVS camera, computed the $r_{on}$ and $r_{off}$ sensory variables and used them to compute the values of $n_1$ and $\omega$ by integrating the two differential equations described above. The value of $\omega$ was then sent by the program to the pan-tilt controller and modified the camera's pan velocity. This implementation illustrates a simple CLP convergence process (**Figure 5**) and shows how different precepts can be differentiated in CLP. The convergence dynamics involves different dynamics of the sensory ($r_{on}$ and $r_{off}$), neuronal ($n_1$, **Figure 6C**) and motor ($\omega$, **Figure 6D**) variables. Yet, the variables are strongly linked, as demonstrated by the phase diagram of the neuronal and motor variables (**Figure 6E**); these two variables quickly converge to a limit cycle (i.e., a constant closed trajectory in the phase plane). Similar behavior is observed in the other phase planes (sensory-motor and sensory-neuronal, not shown). Importantly, in all these phase planes the limit cycle depends on the external contrast ($\gamma$); while maintaining all loop parameters constant, a monotonic change in $\gamma$ results in a corresponding monotonic change of the limit cycle (green and red trajectories in **Figure 6E** for contrasts of 0.5 and 0.9, respectively). Hence, the image's contrast can be inferred from the asymptotic behavior of the system or, in other words, the motor-sensory-neuronal trajectory that is uniquely associated with (or, equivalently, the CLP's IvR of) a given contrast can be "retrieved" by the presentation of that contrast to the perceiver.

The behavior of the SYCLOP is described here in order to demonstrate how a possible implementation of our CLP model would look like. Interestingly, however, it is worth mentioning that the SYCLOP also exhibits behaviors that it was not intentionally designed to exhibit – for example, a smooth pursuit behavior. When presented with a moving image (back and forth horizontal movement of the contrast image at a constant speed) SYCLOP tended to track the image smoothly in each direction (as indicated by the "dwelling spots" at $\omega \approx 5$; and $-5$ $deg/s$ **Figure 6F**).

## Discussion

### Summary of the CLP hypothesis

CLP suggests that perception of the external environment is a process in which the brain temporarily '*grasps*' external objects and incorporates them in its MSM-loops. Such objects become virtual components of the relevant loops, hardly distinguishable, as long as they are perceived, from other components of the loop such as muscles, receptors and neurons. What primarily distinguishes external objects from body parts are inclusion duration and state; short and transient inclusions mark external objects while long and steady inclusions mark body parts (see also *Uexkull, 1926*). Interestingly, the perceptual dynamics suggested by this hypothesis reconciles a conflict between objective scientific observations and the subjective everyday experience of perceiving objects with no mediation (see also *A philosophical angle* below). Everyday perception of a given external object, CLP suggests, is the dynamic process of inclusion of its features in MSM-loops. This process starts with a perturbation, internal or external, and gradually converges towards a complete inclusion - approaching, although never reaching, a state of "direct" perception. A laboratory-induced flashed stimulus, according to this model, probes the initiation of a perceptual process, whereas dreaming and imagining evoke internal components of the process.

### Contrasting OLP and CLP – discriminatory testable predictions

We consider here all versions of OLP, i.e., all versions of hypotheses in which perception does not depend on the integrity of the MSM-loop and its closed-loop dynamics within individual perceptual epochs. We consider here two major OLP classes: in one, sensory OLP (sOLP), the movement of the sensory organ is not an essential component of perception, and in the other, motor-sensory OLP

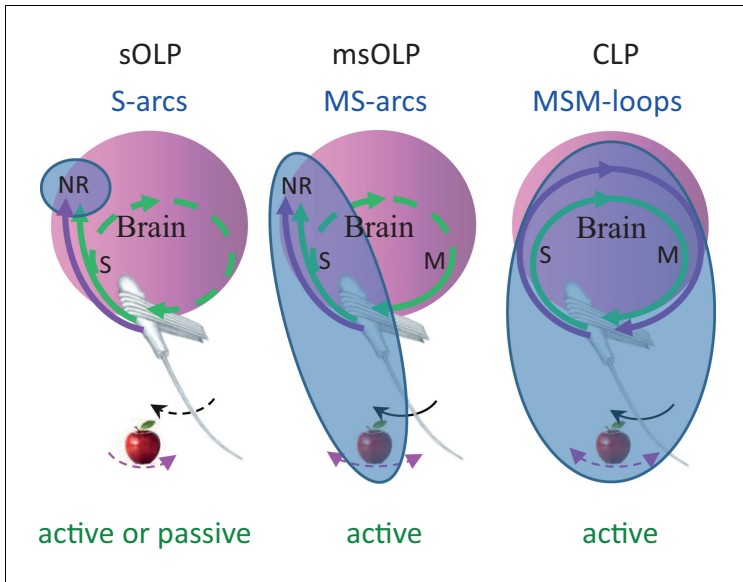

**Figure 7.** Functional connectivity and essential elements of perceptual schemes. The essential elements in each scheme are indicated by solid curves and blue titles.MSM, motor-sensory-motor; MS, motor-sensory; S, sensory; NR, neuronal representation; green curves, re-afferent related pathways; magenta curves, ex-afferent related pathways. Note that re-afferent related pathways can form closed-loops with their sensory organs also in OLP schemes (dashed curves). Arrows indicate optional whisker (black) or object (magenta) movement; solid arrows indicate movements that are essential for perception; in the sOLP scheme none of the movements is essential in itself, but it is essential that at least one of them will occur in order to activate the receptors. Appropriate experimental paradigms are indicated by green titles; CLP and msOLP schemes can be studied only via active sensing paradigms.The minimal sets for invariant representations (IvRs) of external features, i.e., the components that must be included in any IvR according to each perceptual scheme, are marked by the bluish ellipses. sOLP: internal, sensory only NRs. msOLP: sensory NRs + motor-object-sensory contingencies. CLP: entire motor-object-sensory-motor loops.

(msOLP), it is (*Figure 7*). sOLP thus assumes that IvRs are confined to the brain (i.e., they are specific NRs) and can be fully retrieved by sensory activations alone when the sensor is passive. msOLP, in contrast, postulates that IvRs are not confined to the brain, and can form the basis for perception only if they include the relevant MS-contingencies (*Figure 7*). According to msOLP, IvRs cannot be retrieved with passive sensory organs. Importantly, however, msOLP does not assume a motor-sensory-motor loop; that is, its scheme includes a motor-to-sensory arc but not a sensory-to-motor arc (*Figure 7*). Hence, with msOLP, movements of the sensory organ are predetermined for each perceptual epoch and are not affected by the ongoing sensory input during that epoch. In contrast to the OLP hypotheses, using the same representational terminology, CLP postulates that IvRs can form the basis for perception only if they contain the dynamics and state of the entire MSM-loop including the relevant features of the object (*Figure 7*). Thus, the minimal set of variables that must be included in the IvR of each object, or feature, is different for each hypothesis (*Figure 7*, bluish ellipses): internal-only sensory variables in sOLP, internal sensory variables and MS-contingencies in msOLP and the entire perceiving loop in CLP.

Perhaps the first question that comes to mind when considering msOLP and CLP is whether paralyzed subjects perceive stationary (i.e., not flashing or moving) objects similarly to non-paralyzed subjects. If they do - here go the msOLP and CLP hypotheses. Unfortunately, however, this is not a trivial test. Note that the paralysis must include the relevant sensory organ and the object must be entirely stationary. In the case of touch it should be evident that while contacts may be detectable, no object perception is possible with paralyzed hands – we are not aware of any study contradicting this conjecture. In contrast, our intuition regarding hearing is that action is not a fundamental requirement for hearing. Yet, two important points are relevant here. First, our intuition may be misled by the fact that we cannot be aware of motor activation of the outer hair cells and the muscles of the middle ear – we are not aware of perceptual experiments in which these activations were blocked, or measured. Second, no stationary object exists in audition. Acoustic waves are always dynamic and always activate the inner hair cells. This makes auditory sensation less dependent on self-motion, a fact that indeed may put audition in a motor-sensory regime that is distinct from those of touch and vision.

Regarding vision, we are aware of only one study analysing visual perception in a congenital ophthalmoplegic patient, a patient who had no eye movements since birth; in this case, the patient developed a pattern of head movements that resembled that of natural eye movements, only on a slower rhythm (*Gilchrist et al., 1997*). This adaptation clearly indicates the need in active sensation for visual perception, at least in that patient. Natural employment of active vision is indicated by the "weird, confusing and indescribable" forms of perceptions reported during acute partial paralysis of the ocular muscles (*Stevens et al., 1976*). These data are certainly not consistent with sOLP. Yet, these data, as well as part of the OLP-challenging data presented above, may still be consistent with msOLP. The distinction between msOLP and CLP hypotheses is thus more demanding, and requires specifically designed experiments.

We describe here examples of potentially discriminative experiments in three categories.

1. *The motor-to-sensory arc.* The following manipulations are predicted to impair perception according to msOLP or CLP but not according to sOLP: (i) Paralysis of the sensory organ while keeping the sensory flow unimpaired. (ii) Replacing continuous presentation of an object with a series of one or more brief presentations (flashes) while keeping the total stimulus time and/or energy equal.

2. *The sensory-to-motor arc.* The following manipulations are predicted to impair perception according to CLP but not according to sOLP or msOLP: (i) Limiting or forcing sensor movement trajectory via instructions in humans or interventions in rodents. For example, asking humans to scan a scene according to verbal instructions or by pursuing a target, or moving the sensory organ according to a trajectory that was recorded in a previous active session. (ii) Allowing active touch but with the motion of one hand determining the sensory flow to the other hand. (iii) Perturbing neuronal specific sensory-to-motor pathways, such as those connecting the sensory cortex to the motor cortex (*Colechio and Alloway, 2009*), those connecting sensory cortex to motor nuclei (typically via layer 5B neurons), or those connecting the thalamus to motor (cortical and sub-cortical) stations (*Smith et al., 2012*) (*Figure 4A*). The exact design should depend on available genetic markers and the testing of these predictions should be conducted in a balanced way, using appropriate sham perturbations.

The following observations are predicted by CLP but not by sOLP or msOLP during natural

perception: (iv) The motion trajectories of the sensory organ will differ for different object features (expected from affective sensory-to-motor connections). (v) The motion of the sensory organ will depend on the concurrent sensory input; for example, when a rat perceives an object's shape or texture, the movement trajectory of its whiskers will depend on the sequence of curvatures and stick-slip events preceding it within the same perceptual epoch. Similarly, the motion trajectory of the eye will depend on the retinal activations preceding it within the same perceptual epoch.

3. *Motor-sensory-motor convergence.* The following observations are predicted by CLP but not by sOLP or msOLP: (i) The movement trajectory of the sensory organ will show convergence dynamics, i.e., gradual approach to a steady-state pattern, during natural perception. (ii) Convergence will be to different steady-state patterns while perceiving different features or values. (iii) Specific steady-state patterns will be associated with specific perceptual reports. (iv) Convergence dynamics can predict perceptual report timing and/or error. (v) A virtual object can be perceived when sensory neuronal activity is manipulated to mimic the activity expected by the movement of the sensory organ and the presence of a real object (*O'Connor et al., 2013*); with reliable mimicry a convergence process should follow. (vi) CLP predicts that the lag between the actual and perceived times of an external transition ("perceptual lag") should decrease along the process of perceptual convergence, when in steady-state no lag is expected. It has been previously shown that perceptual lags of the onset of transient stimuli are longer than those of continuously-present ones (*Nijhawan, 2001*). CLP thus predicts that with similar experimental protocols (e.g., a rotating arm is shown continuously and a dot is transiently displayed (flashed) for various durations at various positions traversed by the rotating arm), when looking at the offset of the transient stimulus rather than its onset, the temporal perceptual lag of the offset will decrease with increasing transient durations. (vii) With a motion-induced-blindness (MIB) protocol, in which stationary targets are surrounded by moving background dots, the targets 'disappear' occasionally (*Bonneh et al., 2001*). In one possible implementation of CLP the visual system would control the velocity of retinal image slip, and maintain it within a certain working range, instead of directly controlling drift velocity. This would be achieved by modifying drift speed in a manner that is inversely proportional to the speed of the retinal slip. When the retinal slip is dominated by external motion, such as in MIB, eye drift speed would be reduced significantly. When the drift speed will be reduced below a certain level, retinal receptors at corresponding eccentricities may not receive sufficient luminance changes to be activated by the stationary parts of the image. Thus, in MIB conditions in which the drift speed is inversely correlated with the dots' speed, target disappearance is expected to be preceded by a reduction of the drift speed below a certain threshold; threshold level should depend on the eccentricity of the disappearing target.

Ideally, the comparison of the behaviors predicted by CLP and OLP, related to the inter-dependencies of motor, object, sensory and report variables, should be done in natural conditions. Practically, as the scientific method enforces reductionist steps, it is important to notice what reductions are allowed, as behavioral predictions of CLP or msOLP, regarding natural perception, cannot be tested in paradigms in which their basic assumptions are "reduced out." Clearly, if eye or whisker motion is prevented, critical predictions of CLP or msOLP cannot be tested. Experiments in which eye or whisker motion is allowed but head motion is restrained have a limited discriminative power - conclusions in these cases should take into account the possibility that head-restrained animals develop unique compensatory active strategies which may not be indicative for the head-free condition. When MSM-loops are not given enough time to converge, as is the case with passive sensing (e.g., visual flashes) for example, discrimination between CLP and OLP is usually not possible (as both predict partial perception, *Figure 5*).

## A philosophical angle

For at least four centuries the philosophical community, and during the last century also the neuroscience community, have been puzzled by the contrast between objective scientific observations that relate to perception and the everyday subjective experience of perception. What feels direct and immediate to every human perceiver appears indirect and mediated when physical constrains are taken into account (*Crane, 2005*; *Kelso, 1997*; *Port and Van Gelder, 1995*; *Ullman, 1980*). Our CLP hypothesis proposes a reconciliation of objective scientific observations and subjective everyday experience via closed-loop dynamics between the perceiver and the perceived. Such closed-loops converge gradually to a state in which the perceiver and the perceived are inseparable. The idea is

that, although the loops never actually reach an ideal steady-state, they get closer and closer to these states during a perceptual epoch and typically quit the convergence process when the distance from a steady state is barely sensible. Being close enough to the steady state can give rise to the feeling of direct and immediate perception.

In practical terms, this article proposes to open the discussion about the phenomenology and mechanisms of perception, and in particular to confront open- and closed-loop schemes. We hope that the set of predictions listed here will serve as a starting point for informative experimental confrontation.

## Acknowledgements

We thank Merav Ahissar, Amos Arieli, Asher Cohen, Coralie Ebert, Ram Frost, Andrei Gorea, Liron Gruber, Ealan Henis, Rafi Malach, Guy Nelinger, Tess Oram, Kevin O'Regan, Dov Sagi and Avi Saig for helpful comments and discussions and Michal Ahissar for linguistic editing. This work was supported by the Israel Science Foundation (grant #1127/14), the United States-Israel Bi-national Science Foundation (grant #2011432), the NSF-BSF Brain Research EAGER program, (grant #2014906), Israel Ministry of Defense and the Minerva Foundation funded by the Federal German Ministry for Education and Research. EA holds the Helen Diller Family Chair in Neurobiology.

## Additional information

### Funding

| Funder | Grant reference number | Author |
| --- | --- | --- |
| Israel Science Foundation | 1127/14 | Ehud Ahissar |
| United States-Israel Binational Science Foundation | 2011432 | Ehud Ahissar |
| The NSF-BSF Brain Research EAGER program | 2014906 | Ehud Ahissar |
| The Minerva Foundation funded by the Federal German Ministry for Education and Rsearch | | Ehud Ahissar |
| The Israel Ministry of Defense | | Ehud Ahissar |

The funders had no role in study design, data collection and interpretation, or the decision to submit the work for publication.

### Author contributions
EAh, EAs, Contributed to all aspects of this work

### Author ORCIDs
Ehud Ahissar, http://orcid.org/0000-0003-1223-9767

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
