## [Decision Letter]

Thank you for submitting your work entitled "Perception as a closed-loop convergence process" for consideration by *eLife*. Your article has been reviewed by 3 peer reviewers – including the Reviewing editor, David Kleinfeld –, and the evaluation has been overseen by Eve Marder as the Senior Editor. The reviewers have discussed the reviews with one another and the Reviewing Editor has drafted this decision to help you prepare a revised submission. While there was considerable interest and enthusiasm for this work, the reviewers also had some substantive critiques that will require attention.

Summary:

The basic premise of this manuscript is that "…passive paradigms cannot reveal how sensory information is actually processed during active perception […] For that, a unified analysis of the motor and sensory components engaging brains with their environments is required." Toward addressing this premise, "The current paper describes an attempt to bring the motor variables back to theoretical modeling of perception, by proposing a motor-sensory closed-loop scheme for the perception of the external environment." The crux proposal is that internal representation may be a time-varying signal that reaches a steady state behavior perhaps, if only because "Mammalian sensory organs usually acquire information via movements…".

This is a "Viewpoint" with some original material as opposed to an "Original Article" per se. Yet it is timely and important. One would think that most rational neuroscientists would agree. Yet much of mammalian systems neuroscience went through a dark period of working with primarily anesthetized animals or highly constrained animals. This was particularly egregious in vision, where one could argue that a generation of neuroscientists attended to second-order effects to Hubel and Wiesel receptive fields while a basic role of visual areas for motion control went undiscovered until a few years back; see, e.g., Carandini (Nat Neurosci 2013), Bonhoeffer (Neuron 2012), and Stryker (Neuron 2010). The current work reviews this dark period, although the authors should note that some areas, including the study of the VOR and the OKR, the study of hippocampal function during learning and memory (2015 Nobel prize), and clearly the study of motor control for locomotion and manipulation, did not fall into this trap. The pioneering gating experiments of Chapin and Woodward (Exp Neurol 1982), which show how motor output gates sensation, deserve special mention as counterpoint to the authors' sarcasm, i.e., "The (re) discovery that mammalian sensation is active…".

The authors propose that the internal representation of a stimulus depends on motor output. Thus, unless the animal acts on the information, one does not know if the representation, presumably the pattern of neurons spiking in different brain areas, is a complete or only a partial representation. Further, the partial representation could be too incomplete for action to occur. I think we all would agree. Many highly cited studies on internal representation view a change in motor output in response to a change in internal representation – which could be the act of pushing a lever to declare a sensorimotor process is terminating- as a gold standard. There is a fair literature on this – including the pioneering ICMS experiments of Newsome and colleagues (Nature 1990, J Neurophysiol 1992, Neuron 2014). In the vibrissa literature, which appears prominently in this manuscript, there are the reafferent coding studies of Kleinfeld and colleagues (J Neurophysiol 1997; Neuron 2011). The authors go through many arguments to describe why the notion of a motor-free, or open loop representation, will fail. A key argument involves the time it takes – presumably cycles of recurrence – to form an internal representation. This is reminiscent of the argument by Martin (TiNS 1988) on the formation of visual representation as a recurrent of feedback, which was written as a challenge to the feed forward processing implicit in the wiring maps of Feldman and Van Essen.

Essential revisions:

The full reports of all reviewers are appended. All reviewers found merit with the timeliness and importance of the work but all reviewers also found faults that require attention. It is essential to address these issues:

1) Draw a clear distinction about dynamics that spread beyond sensory areas to involve decision making and motor output, each of which may contain local feedback loops, as opposed to brain-wide feedback dynamics per se.

2) Provide clearer and more thoughtful experiments to distinguish between the manifestation of open loop and closed loop representation of the sensory world – at least an object!

3) Properly define and clarify the output from the model / robot (Figure 7).

Specific points:

Reviewer # 1 (annotated by BRE David Kleinfeld):

1) In their paper, “Perception as a closed-loop convergence process”, Ahissar and Assa conceptualize perception as an interactive dynamical process. Specifically the authors propose that perception is a convergence process that involves about 4 repeated sensory-interactions through which an object percept is dynamically generated. Further the authors emphasize the constitutive active nature of sensing and stress the presence of loops rather than of a feed-forward architecture in the brain.

DK: This summary is telling. The reviewer focuses solely on dynamics per se rather than on motor output and control as an integral part of sensation. This implies that the larger message from Ahissar and Assa may have failed to get through.

2) The paper is a strange mix out high-level assumptions and details of rodent active touch. To me these two different levels never fully merged, i.e. it did not become clear to me, where in the rodent brain the dynamical process happens that forms the perceptual object.

DK: In fact, there is published evidence that all of vibrissa L5b cells (in both sensory and motor cortices) have a role in motor control; this goes back to work by Glickman. So this is a clear place to note the origin of a perception and one that is, in terms of hypothesis testing, (just barely) accessible with Ca^2+^-imaging.

3) The predictions that differentiate the OpenLoop and the ClosedLoop model of perception are neither very strong nor very clear. More work is required here.

DK: All reviewers agree on this point. The section on predictions is a crux aspect of the paper that requires significant improvement, as the key is to entice experimentalists to try to falsify or verify the ideas inherent in representation through motor control. This will take thought and time and still may not work out!

4) I have major doubts that the authors are right. It is obvious from the literature that passive, or briefly flashed stimulus presentations, which do not allow active sensing, still evoke robust percepts. I would predict that we will find also a lot more single touch percepts in the active touch system, once we look harder in situations, where animals sensing under time pressure.

DK: I think the confusion results from a mixing of loops for perception, which are hypothesized to include sensory and motor function, and local sensory loops solely for reverberation. The latter are well known to occur with sensory processes, and the most dramatic case is the > 20 s of reverberatory signal in AIT cortex during the delay period of a match to sample task (Fuster & Alexander 1971 Science). Please clarify your text.

Reviewer # 2 (annotated by BRE David Kleinfeld):

1) The distinction between "neural representation" and "internal representation" seems unnecessary – and ill-defined.

DK: This should be fixed.

2) Although I think I understand the intuition behind referring to a sensory-motor loop as "motor-sensory-motor loop", this seems unnecessary; "loop" already implies circularity.

DK: I suspect that this was done to separate loops that are local and lie just in solely sensory or solely regions from brain-wide loops that span the nervous system. As noted above, this needs to be clarified.

3) It is unclear to me what the distinction between msOLP and CLP is?

DK: Please either drop or clarify this issue. It should be a straight forward fix.

4) At times the argument is speculative and unnecessarily strong – to the point of likely already being wrong? E.g., in the subsection “Contrasting OLP and CLP – discriminatory testable predictions” "the question is […] whether paralyzed subjects perceive. If they do – here goes the closed-loop hypothesis". We clearly perceive by hearing without moving. I will grant the authors that it is still unclear what the function of outer hair cells is – but as far as we know, this is a counter example to their hypothesis?

DK: please provide a more graded presentation. The manuscript started out this way, in that open loop representations were a primarily seen as a subset of the larger internal representation.

5) The "mathematical model" and the robotic setup seem to add little to the manuscript. The robotic system example only proves that an oscillatory system can be driven to different attractors with different inputs?

DK: All reviewers commented on the opaque nature of this presentation. It needs to be rewritten. I do not see a fundamental flaw.

6) The testable predictions part is a great idea – but as formulated they are not very helpful. A specific motor output can be thought of as the correlate of a dynamic neural attractor instead of an "instantaneous state".

DK: All reviewers agree. The section on predictions is a crux aspect of the paper that requires significant improvement, as the key is to entice experimentalists to try to falsify or verify the ideas inherent in representation through motor control. This will take thought and time and still may not work out!

Reviewer # 3 (RE Kleinfeld):

1) The statement that "[motor-sensory-motor]-loops are fundamental units of mammalian perception" cannot be right. These loops can support activity and thus a motor-sensory-motor representation, but anatomical loops per se is not a representation.

2) The discussion of two types of loops notes that the "first uses proprioceptive signals […] The other type uses sensory signals to monitor features of external objects…". In fact, this extends confusion in the literature. Signals from muscle spindles, usually regarded as proprioceptive in the sense that are used only for motor feedback, are also sensory. See the pioneering work by Hsiao (2006 J Neurophysiol) on discriminating the size of objects based on muscle stretch.

3) The discussion of reading out the convergence, say, to a limit cycle is muddled. Are convergence cycles related to the accumulation of evidence? If so, it is an interesting idea. But the authors need to describe a mechanism to link dynamics with estimated of confidence. They end with "although the loops never reach the ultimate steady-states, they typically quit the convergence process when the distance from that state is no more sensible". This seems too soft a statement for a serious article.

4) The authors discuss the need to share information between limit cycles (perceptual loops). They are a bit glib in listing possibilities as the locking and unlocking of activity in different loops is essential to their scheme of hierarchical loops. Coherence between different loops is tricky – if the interactions cause a pair of loops to phase-lock, then it is not clear how they separate and dephase. The authors have neglected issues of noise, which is a mechanism to break locking and to dephase.

5) The equations for the SYCLOP model need to be explained. As it stands, this section will lose almost all readers. None of the symbols are explained. I would also start by saying that the simplest model of a loop uses Van der Pol relaxation dynamics. On the one hand it is a bit of a let-down to have the work condensed to a single oscillator that came out of the days of vacuum triodes. On the other hand, the presentation of the realization with the Van der Pol oscillator (Figure 7) is very condensed. I think Figure 7 needs to be considerably unpacked. Panels A, B, and example dynamics like panel H can be one figure, while panels D-G and I could be a second figure. Also, define "k-events", label the ordinates of panels H and I.

6) The authors end with a number of proposed experiments to address the claims of closed versus open loop object representation. One involves the detection of the phase of contact in the whisking cycle, yet is followed by the claim that "…predictions of CLP and OLP can be distinguished only in natural perceiving conditions." This appears to obviate the use of head-fixed animals, an excellent preparation for combined behavior and electrophysiology. Why is head fixing bad for whisking? It seems that perception must often work under partial constraints.

[Editors' note: further revisions were requested prior to acceptance, as described below.]

Thank you for resubmitting your work entitled "Perception as a closed-loop convergence process" for further consideration at *eLife*. Your revised article has been favorably evaluated by Eve Marder (Senior editor), Reviewing editor David Kleinfeld, and one reviewer.

The manuscript has been improved but there are a few remaining issues raised by the reviewer and verified by Reviewing editor Kleinfeld. In order to complete this odyssey, please address these queries.

1) In the subsection “Synthesis of CLP in a robotic setup”. Please expand on the solution of the model, a van der Pol oscillator, to make it transparent to the "typical" biologically trained reader. The statement "The implementation of these equations using the SYCLOP platform" needs to be detailed – even in the appendix – so a reader can duplicate your calculation.

2) In the subsection “Perception can be masked “backwardly”” – "If perception could be reduced to a sequence of pure open loop processes backward masking should not occur." One might think that any slow integration step in a feed-forward processing system would explain backward masking through injection of a signal within the integration time. Perhaps your statement could be better explained as dependent on a system with only "fast" integration.

3) In the subsection “Perception can be masked “backwardly”” – "Perceptual masking thus challenges the validity of the 'virtual knife' reduction and the ability to reconstruct perception based on experiments with flashed stimuli only." The argument leading up to this is not clear.

Finally, please reread the manuscript in a "copy-edit" manner to improve the grammar and correct any number of typos in punctuation.

---

## [Author Response]

The basic premise of this manuscript is that "…passive paradigms cannot reveal how sensory information is actually processed during active perception […] For that, a unified analysis of the motor and sensory components engaging brains with their environments is required." Toward addressing this premise, "The current paper describes an attempt to bring the motor variables back to theoretical modeling of perception, by proposing a motor-sensory closed-loop scheme for the perception of the external environment." The crux proposal is that internal representation may be a time-varying signal that reaches a steady state behavior perhaps, if only because "Mammalian sensory organs usually acquire information via movements…".

This is a "Viewpoint" with some original material as opposed to an "Original Article" per se. Yet it is timely and important. One would think that most rational neuroscientists would agree. Yet much of mammalian systems neuroscience went through a dark period of working with primarily anesthetized animals or highly constrained animals. This was particularly egregious in vision, where one could argue that a generation of neuroscientists attended to second-order effects to Hubel and Wiesel receptive fields while a basic role of visual areas for motion control went undiscovered until a few years back; see, e.g., Carandini (Nat Neurosci 2013), Bonhoeffer (Neuron 2012), and Stryker (Neuron 2010). The current work reviews this dark period, although the authors should note that some areas, including the study of the VOR and the OKR, the study of hippocampal function during learning and memory (2015 Nobel prize), and clearly the study of motor control for locomotion and manipulation, did not fall into this trap. The pioneering gating experiments of Chapin and Woodward (Exp Neurol 1982), which show how motor output gates sensation, deserve special mention as counterpoint to the authors' sarcasm, i.e., "The (re) discovery that mammalian sensation is active…".

Thank you for this concise summary. We would only comment here that our crux proposal is that the perception of external objects is a dynamical process encompassing loops that integrate the organism and its environment and converging towards organism-environment steady-states. We now emphasize this in the Abstract.

The Introduction was revised to cover (and cite) the studies mentioned above and the sarcastic term was removed (subsection “Sensation is normally active”, third paragraph).

The authors propose that the internal representation of a stimulus depends on motor output. Thus, unless the animal acts on the information, one does not know if the representation, presumably the pattern of neurons spiking in different brain areas, is a complete or only a partial representation. Further, the partial representation could be too incomplete for action to occur. I think we all would agree. Many highly cited studies on internal representation view a change in motor output in response to a change in internal representation – which could be the act of pushing a lever to declare a sensorimotor process is terminating- as a gold standard. There is a fair literature on this – including the pioneering ICMS experiments of Newsome and colleagues (Nature 1990, J Neurophysiol 1992, Neuron 2014). In the vibrissa literature, which appears prominently in this manuscript, there are the reafferent coding studies of Kleinfeld and colleagues (J Neurophysiol 1997; Neuron 2011). The authors go through many arguments to describe why the notion of a motor-free, or open loop representation, will fail. A key argument involves the time it takes – presumably cycles of recurrence – to form an internal representation. This is reminiscent of the argument by Martin (TiNS 1988) on the formation of visual representation as a recurrent of feedback, which was written as a challenge to the feed forward processing implicit in the wiring maps of Feldman and Van Essen.

Thank you for these valuable points. We have modified the text to refer to these points and cite the relevant papers – ICMS and confidence (subsection “CLP propositions”), reafference and efference-copy signals (subsection “Perceptual systems are organized as motor-sensory-motor (MSM) loops”) and feedforward versus recurrent processing (subsection “The open loop perception (OLP) doctrine”).

Essential revisions:

The full reports of all reviewers are appended. All reviewers found merit with the timeliness and importance of the work but all reviewers also found faults that require attention. It is essential to address these issues:

1) Draw a clear distinction about dynamics that spread beyond sensory areas to involve decision making and motor output, each of which may contain local feedback loops, as opposed to brain-wide feedback dynamics per se.

This distinction is now clearer. The common dynamics of all components of the relevant MSM-loop(s) is now clearly stated in the Abstract, subsection “CLP propositions”, I, the legend of Figure 5 and the predictions section. The distinction from dynamics of local circuits, as in typical models of decision making, was also added (subsections “The open loop perception (OLP) doctrine” and “CLP propositions”, I).

2) Provide clearer and more thoughtful experiments to distinguish between the manifestation of open loop and closed loop representation of the sensory world – at least an object!

The predictions section was substantially revised. We have categorized the predictions in three groups, focusing on motor-to-sensory, sensory-to-motor, and convergence effects. We have also clarified all predictions, made them more explicit, added examples, and added points raised by the reviewers. In order to emphasize the differences between the predictions of the different schemes we are referring now to the invariant representation (IvR), instead of internal representation (IR), as the representational comparative variable throughout the article. We thank the reviewers for these comments, which significantly helped clarifying our thoughts.

3) Properly define and clarify the output from the model / robot (Figure 7).

The output of the model, as demonstrated by the robot, is now clarified and explained. We modified the description substantially by simplifying the figure (Figure 7, now Figure 6) and extending the text (see below).

Specific points:

Reviewer # 1 (annotated by BRE David Kleinfeld):

1) In their paper, “Perception as a closed-loop convergence process”, Ahissar and Assa conceptualize perception as an interactive dynamical process. Specifically the authors propose that perception is a convergence process that involves about 4 repeated sensory-interactions through which an object percept is dynamically generated. Further the authors emphasize the constitutive active nature of sensing and stress the presence of loops rather than of a feed-forward architecture in the brain.

DK: This summary is telling. The reviewer focuses solely on dynamics per se rather than on motor output and control as an integral part of sensation. This implies that the larger message from Ahissar and Assa may have failed to get through.

We have modified the Abstract to sharpen the crucial suggestions of our hypothesis and make them more explicit, and in particular the crucial role of organism-environment loops. Also, our modifications throughout the paper were done with this issue in mind.

2) The paper is a strange mix out high-level assumptions and details of rodent active touch. To me these two different levels never fully merged, i.e. it did not become clear to me, where in the rodent brain the dynamical process happens that forms the perceptual object.

DK: In fact, there is published evidence that all of vibrissa L5b cells (in both sensory and motor cortices) have a role in motor control; this goes back to work by Glickman. So this is a clear place to note the origin of a perception and one that is, in terms of hypothesis testing, (just barely) accessible with Ca^2+^-imaging.

We probably failed to state it clearly. The dynamical process that forms the perceived object is occurring along the entire MSM-loop(s). Thus, in principle there is no single brain site that preferably represents the object. Still, the comment about L5b makes sense – we now added it to the list of potential tests for the dependency of perception on S-M coupling (subsection “II. The sensory-to-motor arc”). We also make clearer statements about the whole-loop representation in the Abstract and along the paper.

3) The predictions that differentiate the OpenLoop and the ClosedLoop model of perception are neither very strong nor very clear. More work is required here.

DK: All reviewers agree on this point. The section on predictions is a crux aspect of the paper that requires significant improvement, as the key is to entice experimentalists to try to falsify or verify the ideas inherent in representation through motor control. This will take thought and time and still may not work out!

The predictions section was substantially revised. Please see our detailed description in reply to Essential revisions (136) above.

4) I have major doubts that the authors are right. It is obvious from the literature that passive, or briefly flashed stimulus presentations, which do not allow active sensing, still evoke robust percepts. I would predict that we will find also a lot more single touch percepts in the active touch system, once we look harder in situations, where animals sensing under time pressure.

DK: I think the confusion results from a mixing of loops for perception, which are hypothesized to include sensory and motor function, and local sensory loops solely for reverberation. The latter are well known to occur with sensory processes, and the most dramatic case is the > 20 s of reverberatory signal in AIT cortex during the delay period of a match to sample task (Fuster & Alexander 1971 Science). Please clarify your text.

We agree with the reviewer that briefly flashed stimuli evoke robust percepts. However, we argue that this observation is consistent with both OLP and CLP schemes. According to CLP such artificial stimuli initiate the perceptual process, which indeed normally would continue longer and include motor-sensory dynamics but which also can gain information from this initial step (Figure 5 and associated text). What precludes discrimination between OLP and CLP based on flashed stimuli is that although the percepts evoked by flashed stimuli can be robust, they most likely include significantly less information than the information actively acquired from stationary objects. Subjects indeed can differentiate between flashed cars and houses, or animals and humans, but probably cannot perceive the details of the images. We have added a paragraph explaining this in subsection “I. Perception (of external feature(s)) ≡ a process of inclusion in MSM-loop(s)”, fourth paragraph.

We use the terms “most likely” and “probably” because we are not aware of a systematic quantitative comparison of perceptual accuracies of complex images when they are flashed versus being stationary. The first set of predictions in our list includes such a comparison (prediction I-ii; we now added the word “flashes” to make it more explicit) – there we suggest to equalize the total time or energy of the stimuli and thus to use a series of flashes rather than a single one.

Reviewer # 2 (annotated by BRE David Kleinfeld):

1) The distinction between "neural representation" and "internal representation" seems unnecessary – and ill-defined.

DK: This should be fixed.

We have fixed it. We now distinguish between a “Neuronal Representation” (NR), which can be any pattern that shows some correlation with the external feature, and the “Invariant Representations (IvR), which is *the* representation that represents the feature consistently and uniquely – i.e., it occurs *always* and *only* when that feature occurs. This is now explained better in the subsection “The open loop perception (OLP) doctrine”. The minimal sets for IvR according to each perceptual scheme are now described better in Figure 7 (previously Figure 8), its legend and associated text (subsection “Contrasting OLP and CLP – discriminatory testable predictions”).

2) Although I think I understand the intuition behind referring to a sensory-motor loop as "motor-sensory-motor loop", this seems unnecessary; "loop" already implies circularity.

DK: I suspect that this was done to separate loops that are local and lie just in solely sensory or solely regions from brain-wide loops that span the nervous system. As noted above, this needs to be clarified.

Both the criticism and comment are well taken. The major reason for using the term MSM-loop instead of MS-loop is that our repeated experience with presenting the ideas discussed in this paper to colleagues indicated that people often automatically refer to sensory-motor arcs, or to inter-modal sensory-motor loops, when sensory-motor loops are mentioned (see our Figure 4). Thus, many people imagine a kind of a loop that combines visual sensation with arm movement, for example, closing the loop via visual sensation of the arm. This is of course not what we refer to in this paper – we refer to loops that include only one sensory organ, control its movement and sense its signals. We try to emphasize the flow of the loop signals from and to the same sensory organ by using the term motor-sensory-motor loop. We have now expanded the explanation of this term (subsection “subsection “Perceptual systems are organized as motor-sensory-motor (MSM) loops”, third paragraph), which we believe will also help clarifying the scheme we are talking about.

3) It is unclear to me what the distinction between msOLP and CLP is?

DK: Please either drop or clarify this issue. It should be a straight forward fix.

This distinction is important, and we hope that we are now doing a better job in explaining it. The difference is that in msOLP there is no loop – the sensory-to-motor arc is open. We now explain it better in the subsection “Contrasting OLP and CLP – discriminatory testable predictions”, and emphasize it via the classification of our predictions.

4) At times the argument is speculative and unnecessarily strong – to the point of likely already being wrong? E.g., in the subsection “Contrasting OLP and CLP – discriminatory testable predictions” "the question is […] whether paralyzed subjects perceive. If they do – here goes the closed-loop hypothesis". We clearly perceive by hearing without moving. I will grant the authors that it is still unclear what the function of outer hair cells is – but as far as we know, this is a counter example to their hypothesis?

DK: please provide a more graded presentation. The manuscript started out this way, in that open loop representations were a primarily seen as a subset of the larger internal representation.

The auditory case is indeed interesting. We agree with the reviewer that, based on currently available knowledge, it could very well be the case that hearing does not crucially depend on motor outputs. Yet, the experiment had not been yet done – one needs to paralyze or block the outputs to the outer hair cells and the muscles of the middle ear in order to test it. We now state it in the second paragraph of the subsection “Contrasting OLP and CLP – discriminatory testable predictions”.

There is an additional point here. The auditory stimulus is fundamentally different than the visual and tactile ones – it can never be stationary. There is no stationary acoustic wave. Thus, the inner hair cells are always activated by a sound. This makes the dependence on sensor motion less crucial. Thus, we have modified the question to “whether paralyzed subjects perceive stationary (i.e., not flashing or moving) objects similarly to non-paralyzed subjects” (see aforementioned paragraph). Given the reviewer’s comment we also found it important to elaborate further on the special case of auditory sensation.

5) The "mathematical model" and the robotic setup seem to add little to the manuscript. The robotic system example only proves that an oscillatory system can be driven to different attractors with different inputs?

DK: All reviewers commented on the opaque nature of this presentation. It needs to be rewritten. I do not see a fundamental flaw.

We agree with the criticism – the description was too laconic and encrypted. We modified this section substantially by simplifying the figure (Figure 7, now Figure 6 – removing the open-loop responses) and explaining the outcome of the robotic model and its significance (subsection “Synthesis of CLP in a robotic setup”). We think that the demonstrations of how a convergence process may look like, and how a steady-state may look like, are of value in this paper as they can help the reader capturing the type of processes we refer to – we thus prefer to leave this section in.

6) The testable predictions part is a great idea – but as formulated they are not very helpful. A specific motor output can be thought of as the correlate of a dynamic neural attractor instead of an "instantaneous state".

DK: All reviewers agree. The section on predictions is a crux aspect of the paper that requires significant improvement, as the key is to entice experimentalists to try to falsify or verify the ideas inherent in representation through motor control. This will take thought and time and still may not work out!

The predictions section was substantially revised. Please see our detailed description in reply to Essential revisions (2) above.

Reviewer # 3 (BRE David Kleinfeld):

1) The statement that "[motor-sensory-motor]-loops are fundamental units of mammalian perception" cannot be right. These loops can support activity and thus a motor-sensory-motor representation, but anatomical loops per se is not a representation.

Good point. Indeed, whenever we refer to MSM loops we refer to both their anatomical and functional levels. We are now stating this explicitly in the subsection “Perception can be masked “backwardly”.

2) The discussion of two types of loops notes that the "first uses proprioceptive signals […] The other type uses sensory signals to monitor features of external objects…". In fact, this extends confusion in the literature. Signals from muscle spindles, usually regarded as proprioceptive in the sense that are used only for motor feedback, are also sensory. See the pioneering work by Hsiao (2006 J Neurophysiol) on discriminating the size of objects based on muscle stretch.

Another good point – thank you! Indeed, proprioceptive loops can provide (limited) information about external objects. For example, when a large error between the planned and executed movement occurs, an external object that blocks the movement is a natural interpretation of the brain. We now added this point and a citation of Hsiao’s paper in the subsection “The closed-loop perception (CLP) hypothesis”, last paragraph.

3) The discussion of reading out the convergence, say, to a limit cycle is muddled. Are convergence cycles related to the accumulation of evidence? If so, it is an interesting idea. But the authors need to describe a mechanism to link dynamics with estimated of confidence. They end with "although the loops never reach the ultimate steady-states, they typically quit the convergence process when the distance from that state is no more sensible". This seems too soft a statement for a serious article.

In general we consider the formalization of the link between dynamics and confidence to lie outside the scope of the current paper. Yet, we agree that the description of possible mechanisms is in place here. We have thus added a paragraph describing the potential mechanism we prefer, which is based on internal models (subsection “I Perception (of external feature(s)) ≡ a process of inclusion in MSM-loop(s)”, fifth paragraph).

4) The authors discuss the need to share information between limit cycles (perceptual loops). They are a bit glib in listing possibilities as the locking and unlocking of activity in different loops is essential to their scheme of hierarchical loops. Coherence between different loops is tricky – if the interactions cause a pair of loops to phase-lock, then it is not clear how they separate and dephase. The authors have neglected issues of noise, which is a mechanism to break locking and to dephase.

We assume here that the loops composing an object perception remain engaged until the perceptual epoch ends. In this case dephasing will automatically follow. We have added a sentence mentioning that in the first paragraph of the subsection “II. Perception of an external object ≡ a coordinated process of inclusion in a collection of MSM-loops”.

5) The equations for the SYCLOP model need to be explained. As it stands, this section will lose almost all readers. None of the symbols are explained. I would also start by saying that the simplest model of a loop uses Van der Pol relaxation dynamics. On the one hand it is a bit of a let-down to have the work condensed to a single oscillator that came out of the days of vacuum triodes. On the other hand, the presentation of the realization with the Van der Pol oscillator (Figure 7) is very condensed. I think Figure 7 needs to be considerably unpacked. Panels A, B, and example dynamics like panel H can be one figure, while panels D-G and I could be a second figure. Also, define "k-events", label the ordinates of panels H and I.

The equations of the SYCLOP model are now better explained (subsection “Synthesis of CLP in a robotic setup”). The outcome of SYCLOP model is now described in detail and the motivation for using Van der Pol dynamics is now explained as well (in the aforementioned subsection).

Figure 7 (now Figure 6) was substantially simplified – the open-loop response panels were removed and the figure now conveys the main messages in a clearer manner.

6) The authors end with a number of proposed experiments to address the claims of closed versus open loop object representation. One involves the detection of the phase of contact in the whisking cycle, yet is followed by the claim that "…predictions of CLP and OLP can be distinguished only in natural perceiving conditions." This appears to obviate the use of head-fixed animals, an excellent preparation for combined behavior and electrophysiology. Why is head fixing bad for whisking? It seems that perception must often work under partial constraints.

We agree. The scope of the term “natural conditions” is too wide. We have modified this paragraph substantially and now explain better which reductionist paradigms would not allow a meaningful testing of the hypotheses. We also now explain better in that paragraph how different conditions, including head fixation, should be taken into account (subsection “III. Motor-sensory-motor convergence”, last paragraph).

[Editors' note: further revisions were requested prior to acceptance, as described below.]

The manuscript has been improved but there are a few remaining issues raised by the reviewer and verified by Reviewing editor Kleinfeld. In order to complete this odyssey, please address these queries.

1) In the subsection “Synthesis of CLP in a robotic setup”. Please expand on the solution of the model, a van der Pol oscillator, to make it transparent to the "typical" biologically trained reader. The statement "The implementation of these equations using the SYCLOP platform" needs to be detailed – even in the appendix – so a reader can duplicate your calculation.

We have expanded on the solution of the model, and explained the choice of the van der Pol oscillator (subsection “Synthesis of CLP in a robotic setup”). We also provide more details on the SYCLOP implementation, and explain how the model equations were implemented (in the aforementioned subsection).

2) In the subsection “Perception can be masked “backwardly”” – "If perception could be reduced to a sequence of pure open loop processes backward masking should not occur." One might think that any slow integration step in a feed-forward processing system would explain backward masking through injection of a signal within the integration time. Perhaps your statement could be better explained as dependent on a system with only "fast" integration.

The section on backward masking is now clearer. We explain the ‘standard model’ for open loop schemes, which is based on dual channel (fast and slow) integration and interaction (Breitmeyer & Ogmen’s 2000, reference added), its inconsistency with experimental data, and the challenge it forms for OLP (subsection “Perception can be masked “backwardly”).

3) In the subsection “Perception can be masked “backwardly”” – "Perceptual masking thus challenges the validity of the 'virtual knife' reduction and the ability to reconstruct perception based on experiments with flashed stimuli only." The argument leading up to this is not clear.

We hope that the improved explanation of the backward masking challenge (point 2) provides a better background for understanding the statement about the virtual knife. In addition, we have modified this sentence to be more explicit and clear (subsection “Perception can be masked “backwardly”). A related change was introduced in the fourth paragraph of the subsection “I Perception (of external feature(s)) ≡ a process of inclusion in MSM-loop(s)”.

Finally, please reread the manuscript in a "copy-edit" manner to improve the grammar and correct any number of typos in punctuation.

We have reread the entire paper, fixed typos and grammatical mistakes (with the aid of a linguistic editor) and improved the clarity of the text where needed – thanks for noting that. Figure 1 and Figure 7 were slightly modified (Figure 1: eye – object arrows added; Figure 7: graphics).